# Understanding Mistakes in Transformers through Token-Level Semantic Dependencies

## Abstract

Despite the high performance of the transformer model, it sometimes produces incorrect information. To understand the cause of this issue, we explore how semantic dependency is learned within the model. Specifically, we investigate how tokens in multi-head self-attention transformer models encode semantically dependent information. To help us identify the semantic information encoded within a token, intuitively, our method analyzes how a token's value shifts in response to changes in semantics. BERT, LLaMA and GPT models are analyzed. We have observed some interesting and similar behaviors in their mechanisms for encoding semantically dependent information: 1). Most tokens primarily retain their original semantic information, even as they pass through multiple layers. 2). A token in the final layer usually encodes truthful semantic dependencies. 3). The semantic dependency within a token is sensitive to both irrelevant context changes and order of contexts. 4). Mistakes made by the model can be attributed to some tokens that falsely encode semantic dependencies. Our findings potentially can help develop more robust and accurate transformer models by pinpointing the mechanisms behind semantic encoding.

## 1 Introduction

Transformer models have revolutionized the field of natural language processing (NLP) since their introduction by (Vaswani et al., 2017). By leveraging self-attention mechanisms, transformers enable models to capture long-range dependencies in text, leading to significant advancements in tasks such as machine translation, text summarization, and language generation. Popular language models such as BERT (Devlin et al., 2018), the GPT series (Radford et al., 2019; Brown, 2020), and LLaMA (Touvron et al., 2023) are based on the transformer architecture and have set new benchmarks. They showcase the transformer's capacity to understand and generate human-like text.

Large Language Models (LLMs) have demonstrated remarkable capabilities across various natural language tasks. However, alongside their benefits, LLMs pose significant risks and challenges (Weidinger et al., 2021). Research has shown that LLMs may intensify biases in training data (Navigli et al., 2023; Taori & Hashimoto, 2023), produce toxic content (Gehman et al., 2020; Ousidhoum et al., 2021), generate false information (Lin et al., 2021), and exhibit hallucinations (Ji et al., 2023). Additionally, concerns have been raised about LLMs leaking sensitive training data (Carlini et al., 2021) and engaging in deceptive behaviors (OpenAI, 2023; Scheurer et al., 2024). Addressing these issues has led to the development of evaluation methods for LLM performance (Liang et al., 2022) and strategies aimed at mitigating harmful outputs. (Ganguli et al., 2022; Bai et al., 2022).

Existing research has elucidated several reasons contributing to the errors observed in LLMs. Studies have suggested that non-linearity, insufficient model averaging, and inadequate regularization in deep learning models lead to mistakes when encountering crafted adversarial examples (Chakraborty et al., 2018; Zhang et al., 2020). Additionally, Kang et al. (2024) indicated that the programmatic behavior of LLMs may result in vulnerabilities under security attacks, leading to the generation of harmful content. Wei et al. (2024) attribute the susceptibility of safety-trained LLMs to competing objectives and mismatched generalization. Extensive studies also indicate various reasons for language models generating unfaithful or nonsensical text, including source-reference divergence in data, imperfect representation learning, erroneous decoding, exposure bias, and parametric knowledge bias (Ji et al., 2023).

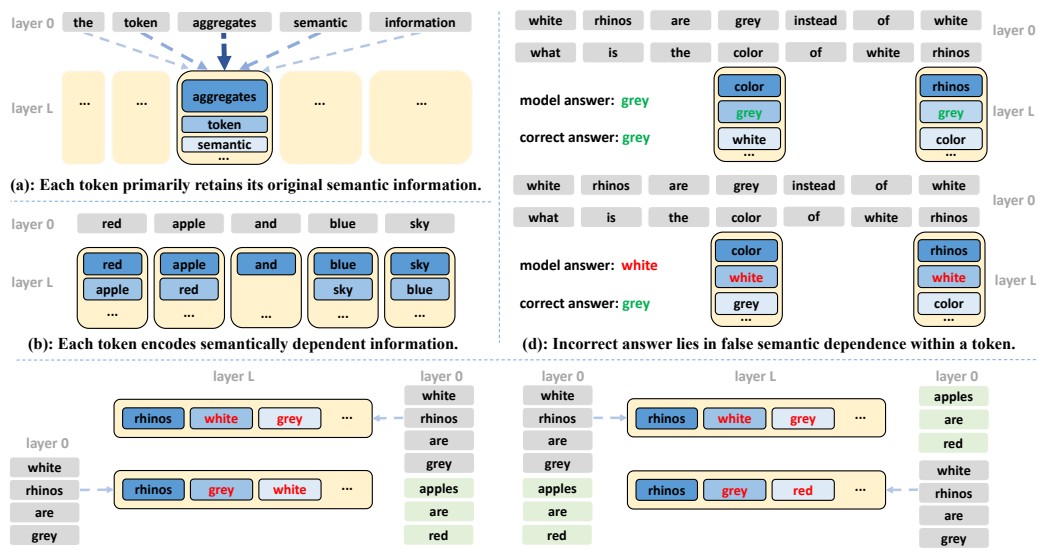

Figure 1: An illustration of our key findings regarding the behavior of tokens in semantic information aggregation and propagation. Different transformer models (i.e., BERT, LLaMA, and GPT) are used.

These studies have identified various reasons that lead to errors and have enhanced our understanding, providing valuable insights into model weaknesses. Building upon these insights, we aim to delve deeper into the internal mechanisms within the model's architecture that lead to errors. We believe that errors produced by LLMs can arise from the way semantic information is propagated and aggregated across tokens within transformer models.

Semantic information refers to the meaningful content that consists of data or representations that carry meaning interpretable in a specific context. Semantic dependency can be defined as the relationship between words in a sentence where the meaning of one word (predicate) depends on another word (argument) in the sentence (Mel'čuk, 2001). In our case, false semantic dependency means the meaning of one word is not dependent on another. For example, in sequence "blue sky and red apple", the semantic dependency between word "blue" and "apple" are false. Our intuition is that in transformer models, inputs are tokenized and embedded into vectors representing semantic information. These tokens are then processed through multiple attention layers, where semantic information is propagated between tokens in each layer. This process enables the model to build semantic dependency for generating coherent and contextually relevant outputs. However, inaccuracies in this propagation process can lead to errors in the model's predictions. Errors in LLMs outputs typically manifest as incorrect probability predictions in the final layer. These predictions rely heavily on the token representations produced by the preceding layers. Therefore, it is plausible that such errors stem from incorrect propagation or misinterpretation of semantic information across tokens during the forward pass. Misalignment in semantic information can disrupt the model's "contextual understanding", leading to the generation of inaccurate outputs.

To systematically explore how semantic information is propagated and aggregated within transformer models, our objective is how tokens within transformer models propagate and encode semantic information. We propose methods to interpret the information aggregation mechanisms of transformer models. The philosophy is that when an input token carrying semantic information is altered, the tokens that receive this information through the transformer will exhibit significant changes in their outputs, while irrelevant tokens remain relatively unchanged. Therefore, by evaluating the variation in output tokens when introducing perturbations in the input tokens, we can track the aggregation of semantic information related to various concepts in token representations.

**Key Findings** In our exploration, we analyzed different transformer models such as BERT, LLaMA and GPT. We discovered several key findings regarding the behavior of tokens for semantic information aggregation and propagation. Each finding provides insights into how these models aggregate and propagate semantic information, which could be important for future model design.

1). We found that **most tokens primarily retain their original semantic information, even as they pass through the layers of transformers**. For example, in Figure 1(a), the arrows indicate the semantic information flow from the token at layer 0 to token at layer L. For the token "*aggregates*" in the input token sequence in layer 0, the final layer's token aggregates a large amount of information from its input token and a small amount of information from other tokens. The fact that most tokens still predominantly reflect their initial semantics highlights model's strong retention property, which is not inherently expected given the iterative aggregation of semantic information across many layers.

2). **We found that a token in the final layer usually encodes truthful *semantic dependency*.** Note that semantic dependency refers to the relationship between words in a sentence where the meaning of one word depends on another word in the sentence. In the case of the input "red apple and blue sky" shown in Figure 1(b), an output token will encode the semantically **dependent** information "red" and "apple" together, rather than encoding semantically **independent** information like "blue" and "apple". Therefore, it usually encodes truthful semantic dependency.

3). We found that **The encoded semantic dependency within a token is influenced by both irrelevant context changes and the order of contexts.**. For example, we have two semantically independent token sequences "*white rhinos are gray*" and "*apples are red*" in Figure 1(c), where "apples are red" serves as irrelevant context to "white rhinos are gray." On the left side of the figure, when we add the irrelevant context "apples are red", the rank of semantic dependency strength between the token "rhinos" and tokens in its sequence "white rhinos are gray." varied. On the right side of the figure, the same thing happens when we maintain the overall input semantic information unchanged and only change the order of the two token sequences. This demonstrates that even when two token sequences are semantically independent, irrelevant changes in context and the ordering of sequences can significantly alter how semantic information is aggregated within each token.

4). The above three findings serve as prerequisite of studying how token-level semantic dependency influences model mistakes. Finally, we found that **when the model makes mistakes, certain tokens erroneously encode information that should not exhibit semantic dependency**. For example, Figure 1(d) demonstrates that semantic information is aggregated differently in the output token sequence when the model outputs an incorrect answer. In a question-answering task where the context sequence "*white rhinos are grey instead of white*" is paired with the question "*What is the color of white rhinos?*", the correct answer is "*grey*". However, when the model incorrectly outputs "*white*", the question's key terms, such as "color" and "rhinos", contain more information about "*white*" rather than "*grey*". This highlights how false relationships between key tokens can lead to incorrect outputs.

**Implications for Future Model Design** Our insights into semantic information propagation and aggregation within tokens of transformer models potentially help design new transformer architectures to be more resilient and semantically coherent. For example, our third finding demonstrates that irrelevant context and context order significantly influence the semantic dependencies within tokens. A natural thought for future work could be on regulating transformer models to maintain consistent semantic dependencies despite irrelevant context variations. This may involve implementing regularization techniques that enforce stable token representations regardless of irrelevant context or sequence alterations. As another example, our fourth finding reveals that model errors often result from certain tokens erroneously encoding semantic dependencies that should not exist. To address this, future research could refine attention mechanisms to better prioritize meaningful token interactions and reduce the impact of adversarial context. This could be achieved by implementing dynamic reweighting strategies in attention heads and incorporating stricter regularization techniques can prevent tokens from erroneously encoding unrelated information.

## 2 MOST TOKENS PRIMARILY RETAIN THEIR ORIGINAL SEMANTIC INFORMATION THROUGH TRANSFORMER LAYERS

In this section, we investigate how individual tokens propagate semantic information through the layers of transformer models. We find that 1). through the transformer's layers, most final-layer tokens still primarily maintain their original semantic information; 2). each final-layer token contains varying levels of semantic information from the entire sequence.

**Transformer Architecture**  We consider a general $L$-layer transformer model. Each layer consists of a multi-head self-attention mechanism (MHA) followed by a position-wise feed-forward network (FFN), along with residual connections. The input sequence of $N$ tokens is embedded into $D$-dimensional vectors and combined with positional encodings to form the initial representations:

$$\mathbf{z}^0 = [\mathbf{z}_1^0, \mathbf{z}_2^0, \ldots, \mathbf{z}_N^0], \tag{1}$$

where $\mathbf{z}_i^0 \in \mathbb{R}^D$ is the embedding of the $i$-th token in layer 0.

In transformer-based models, the token sequence is updated through $L$ layers using the following two steps, where multi-head attention (MHA) and feed-forward networks (FFN) work together to enrich the text representations:

$$\hat{\mathbf{z}}^l = \text{MHA}^l(\mathbf{z}^{l-1}) + \mathbf{z}^{l-1}, \quad \mathbf{z}^l = \text{FFN}^l(\hat{\mathbf{z}}^l) + \hat{\mathbf{z}}^l, \tag{2}$$

where $l = 1, 2, \ldots, L$. Here, $\text{MHA}^l$ and $\text{FFN}^l$ denote the multi-head attention and feed-forward network operations at layer $l$, respectively. The residual connections ensure that information flows directly through layers, facilitating the retention of original semantic information. For the $i$-th token in the output of the $L$-th layer, we have:

$$\mathbf{z}_i^L = \mathbf{z}_i^0 + \sum_{l=1}^{L} \text{MHA}_i^l(\mathbf{z}^{l-1}) + \sum_{l=1}^{L} \text{FFN}_i^l(\mathbf{z}^l), \tag{3}$$

where $\text{MHA}_i^l$ and $\text{FFN}_i^l$ represent the operations affecting the $i$-th token at layer $l$ (Vaswani et al., 2017). Note that the above equation is used to show that a last-layer token can be written as a combination of first-layer tokens. We use the formulation proposed in Gandelsman et al. (2024), which ignores the layer-normalization term.

To validate that the $i$-th token $\mathbf{z}_i^L$ in the output layer $L$ primarily contains information about the $i$-th token in the input layer $\mathbf{z}_i^0$, we compare the changes of all tokens in the final layer $L$ with the changes in $\mathbf{z}_i^0$. The key idea is that if the token $\mathbf{z}_i^L$ is the most affected when the token $\mathbf{z}_i^0$ changes, it indicates that removing the information in $\mathbf{z}_i^0$ by altering $\mathbf{z}_i^0$ leads to the most significant change in $\mathbf{z}_i^L$. This suggests that the $i$-th token in the final layer encodes most information derived from the $i$-th token in the first layer.

**Token Perturbation**  We then generate $K$ perturbed versions of the input token $\mathbf{z}^{0(\text{org})}$ by only replacing the $i$-th token $\mathbf{z}_i^0$ with randomly sampled tokens from the vocabulary $\mathcal{V}$. Specifically, we sample a new token $\tilde{\mathbf{z}}_i^{0(k)}$ for $k$ times as follows.

original $\mathbf{z}^{0(\text{org})} = [\mathbf{z}_1^0, \ldots, \mathbf{z}_i^0, \ldots, \mathbf{z}_N^0]$; perturbed $\tilde{\mathbf{z}}^{0(k)} = [\mathbf{z}_1^0, \ldots, \tilde{\mathbf{z}}_i^{0(k)}, \ldots, \mathbf{z}_N^0]$,

where $\tilde{\mathbf{z}}_i^{0(k)} \sim \text{Uniform}(\mathcal{V})$  and $k \in \{1, \ldots, K\}$. $\tag{4}$

Each perturbed sequence of token $\tilde{\mathbf{z}}^{0(k)}$ is processed independently through the L-layer transformer model, yielding L-layer token $\tilde{\mathbf{z}}^{L(k)}$. Similar the corresponding L-layer token for $\mathbf{z}^{0(\text{org})}$ is $\mathbf{z}^{L(\text{org})}$.

**Measuring Semantic Information Dependency**  To quantify how the perturbation of the $i$-th token $\mathbf{z}_i^0$ in the first layer affects $j$-th token $\mathbf{z}_j^0$ in final layer, we examine the average change of the $j$-th token across the $K$ sequences. Specifically, for the $j$-th token, we calculate the semantic dependency score $\Delta_{\mathbf{z}_j^L | \mathbf{z}_i^0}$, which is achieved by calculating average change $\Delta_{\mathbf{z}_j^L | \mathbf{z}_i^0}$ between its value in the original sequence and its values in the perturbed sequences:

$$\Delta_{\mathbf{z}_j^L | \mathbf{z}_i^0} = \frac{1}{K} \sum_{k=1}^{K} \left\| \tilde{\mathbf{z}}_j^{L(k)} - \mathbf{z}_j^{L(\text{org})} \right\|_2. \tag{5}$$

A higher value of $\Delta_{\mathbf{z}_j^L | \mathbf{z}_i^0}$ indicates that the $j$-th token in final layer $L$ is more sensitive to change of the $i$-th token. It implies that $j$-th token should encode more information from the $i$-th token.

Table 1: Percentage of a token primarily retains its original semantic information.

| | gsm8k | Yelp | GLUE | DailyMail | OpenOrca | WikiText |
|---|---|---|---|---|---|---|
| BERT | 99.22 | 98.58 | 98.48 | 98.81 | 98.90 | 98.84 |
| RoBERTa | 92.29 | 95.16 | 94.43 | 95.11 | 94.38 | 94.39 |
| ALBERT | 96.84 | 97.36 | 97.67 | 96.67 | 97.65 | 95.85 |
| DistilBERT | 93.84 | 95.27 | 95.84 | 95.70 | 95.54 | 94.49 |
| GPT-2 | 75.19/88.42 | 77.46/89.94 | 77.49/92.51 | 73.11/85.88 | 69.32/81.68 | 72.31/84.46 |
| LLama3 | 96.21 | 96.68 | 94.20 | 95.85 | 95.78 | 94.80 |

To validate that the $j$-th token $\mathbf{z}_j^L$ in the output layer $L$ primarily contains information about the $i$-th token in the input layer $\mathbf{z}_i^0$, we compare the average change $\Delta_{\mathbf{z}_j^L | \mathbf{z}_i^0}$ for all tokens $j \in \{1, \dots, N\}$. We check whether the average change $\Delta_{\mathbf{z}_i^L | \mathbf{z}_i^0}$ is the largest among all $\Delta_{\mathbf{z}_j^L | \mathbf{z}_i^0}$, indicating that perturbing the $i$-th token affects its own output token more than any other token's output. By comparing the $\Delta_{\mathbf{z}_j^L | \mathbf{z}_i^0}$ values for all tokens $j \in \{1, \dots, N\}$, we can determine which token in the final layer encode most information about $i$-th token. To quantify this observation across multiple instances, we calculate the percentage $P$ that the $i$-th token's perturbation in output layer primarily affects its corresponding output token in a transformer-based language model $f_\theta$ on $M$ tested token cases as follows:

$$P(f_\theta) = \frac{1}{M} \sum_{m=1}^{M} \mathbb{1}_{\{i = \arg\max_j^N \Delta_{\mathbf{z}_j^L | \mathbf{z}_i^0}\}}.$$

**Experiment** We measure the total percentage with various sentences from six datasets, including gsm8k (Cobbe et al., 2021), Yelp (Zhang et al., 2015), GLUE (Wang et al., 2019), CNN/DailyMail (Hermann et al., 2015), OpenOrca (Lian et al., 2023) and WikiText (Merity et al., 2016). For each model, over 100,000 token cases were evaluated across datasets (each token perturbation is treated as one case). Noted that we compute changes for nearly all tokens (over 95%) in each sequence, excluding special tokens such as [CLS] and [SEP],which ensures a comprehensive assessment of the semantic dependency across the input. The results, displayed in Table 1, show the percentage that a token primarily retains its original semantic information.

Our Experiment compare models including BERT series (encoder only), GPT(decoder-only, autoregressive) and Llama(decoder-only, auto-regressive). Compared to BERT and LLama, there is a part of tokens that does not preliminary retain its original information in GPT. We also include the percentage of the token propagate semantic information to both of its next token and itself (shown in Table 1). From this experiment, we can conclude that most tokens primarily retain their original semantic information, even as they pass through the transformer layers. Additionally, we also observe that the influence of each input token on other output tokens in the final layer exists almost 100%.

## 3 A FINAL-LAYER TOKEN ENCODES TRUTHFUL SEMANTIC DEPENDENCY

In the previous section, we observed that most tokens primarily retain their original semantic information even they propagate through the transformer layers. However, we also found that perturbing a specific input token can cause variations in the outputs of other tokens in the final layer. This suggests that tokens not only retain their own semantic information but also integrate semantic information from all other tokens. In this section, we aim to verify whether a token usually contains semantically dependent information. Specifically, we investigate if tokens encode more semantic information from semantically related words compared to unrelated words in the sequence. We find that this holds for most tokens.

To check whether tokens effectively encode semantically dependent information, we first randomly select a word, denoted as $\mathbf{w}_i^0$. We then identify a group $G_{\mathbf{z}_i^0}$ containing the indices of semantically dependent tokens by leveraging semantic dependency parsing tools SpaCy (Honnibal et al., 2020), which parse the words in the sentence that are semantically dependent with $\mathbf{w}_i^0$, including both head and children in parsing tree and the word itself. Spacy works by using a pre-trained neural network model to predict the syntactic relationships between tokens, which provided than human annotations. Next, we estimate $\hat{G}_{\mathbf{z}_i^0}$ by changing $\mathbf{z}_i^0$ and obtain the indices of top $K_{\text{top}}$ tokens that most sensitive to the change of $\mathbf{z}_i^0$. Finally, we calculate the average similarity between these two sets.

**Semantically Dependent Token Groups** A group $G_{\mathbf{z}_i^0}$ containing the indices of semantically dependent tokens with $\mathbf{z}_i^0$. To identify a semantically dependent token group $G_{\mathbf{z}_i^0}$, we can leverage semantic dependency parsing methods to get the semantic word group $W_{\mathbf{w}_i^0}$, then convert it to a token group. Intuitively, dependency parsing analyzes the grammatical structure of a sentence, establishing relationships between *"head"* words and the words that modify them. For example, in the sentence *"The quick brown fox jumps over the lazy dog."*, the word *"fox"* is semantically related to word *"quick"*, *"brown"* and *"jumps"* based on their grammatical dependencies.

Given the semantic word group $W_{\mathbf{z}_i^0}$ by using existing semantically dependency parsing methods. Once the semantic word group $W_{\mathbf{w}_i^0}$ of the word $\mathbf{w}_i^0$ is identified, each word $\mathbf{w}_j$ in $W_{\mathbf{w}_i^0}$ is converted into its corresponding token indices, and $\mathbf{w}_i^0$ also is converted into $\mathbf{z}_i^0$, which obtains $G_{\mathbf{z}_i^0}$ [1].

**Estimated Semantically Dependent Token Group by Leveraging Token Perturbation** To estimate the semantically dependent word group $\hat{G}_{\mathbf{z}_i^0}$ for each token $\mathbf{z}_i^0$, we measure semantic information propagation $\Delta_{\mathbf{z}_j^L | \mathbf{z}_i^0}$ by Eq. (7) for each token $\mathbf{z}_j^L$ in the final layer $L$. Then we rank it and select the largest $K_{\text{top}}$ indices within the sequence into a set denoted as $\hat{G}_{\mathbf{z}_i^0}$.

$$\hat{G}_{\mathbf{z}_i^0} = \{j \mid j \in \text{indices of max}_{K_{\text{top}}}(\Delta_{\mathbf{z}_j^L | \mathbf{z}_i^0}, j = 1, \dots, N)\}. \tag{6}$$

**Calculating Alignment Score** To assess the alignment between the most affected tokens and the semantically related word group $G_{\mathbf{z}_i^0}$, we compute the alignment score $S_i$ to measure the overlap between $\hat{G}_{\mathbf{z}_i^0}$ and $G_{\mathbf{z}_i^0}$:

$$S_{\mathbf{z}_i^0} = \frac{\left| G_{\mathbf{z}_i^0} \cap \hat{G}_{\mathbf{z}_i^0} \right|}{K_{\text{top}}}, \tag{7}$$

where $\left| G_{\mathbf{z}_i^0} \cap \hat{G}_{\mathbf{z}_i^0} \right|$ represents the number of overlapping tokens between $G_{\mathbf{z}_i^0}$ and $\hat{G}_{\mathbf{z}_i^0}$.

**Experiment** We conducted this experiment on several transformer models, including BERT, RoBERTa, ALBERT, Distil-BERT, Llama3, and GPT-2. We firstly construct a specialized word dependency dataset using SpaCy. This dataset includes sentences from the GLUE dataset, where each word (as one case) in the sentence is annotated with its semantically dependent word groups as standard dependency data. For each model, we evaluated over 10,000 cases, where each case corresponds to perturbing a single token and computing the alignment score. These results demonstrate that the tokens most affected by the perturbation of $\mathbf{z}_i^0$ tend to be the ones that are semantically related to it. This indicates that tokens particularly integrate semantic information from semantically dependent tokens.

Table 2: Alignment scores indicate how well individual tokens encodes truthful semantic dependency (%).

| Model | Alignment Score (%) |
|---|---|
| BERT | 87.86 |
| RoBERTa | 82.44 |
| ALBERT | 88.77 |
| DistilBERT | 88.88 |
| GPT-2 | 93.41 |
| Llama3 | 92.47 |

The averaged alignment scores across all cases are presented in Table 2. The overall high alignment scores across different models, which demonstrates that our method effectively captures the semantic dependencies between tokens.

## 4 THE SEMANTIC DEPENDENCY ENCODED IN A TOKEN IS INFLUENCED BY BOTH IRRELEVANT CONTEXT CHANGES AND ORDER OF CONTEXTS

Intuitively, semantic dependencies between tokens should remain robust regardless of changes in *irrelevant* context or the order of independent sentences. We would like to know how the existing transformer model behaves. Motivated by this curiosity, we conducted an experiment to determine whether altering the irrelevant context or rearranging the order of independent sentences affects established semantic dependencies.

---

[1] In our experiments, we do not consider the case when $\mathbf{w}_i^0$ is converted to subword tokens.

**Semantic Dependency Analysis with Irrelevant Context Change**   To validate whether irrelevant context influences the semantic dependencies of tokens in a sequence, we selected two semantically independent sentences randomly sampled from a dataset. Consider two sentences:

*"The sky is blue."* vs *"The apple is red. The sky is blue.",* i.e., $s_1$ vs $(s_2, s_1)$

*"The sky is blue."* vs *"The sky is blue. The apple is red.",* i.e., $s_1$ vs $(s_1, s_2)$

We investigated whether the semantic dependencies within "*The sky is blue.*" remain unchanged when appended with "The apple is red." on its left side or right side. Since both contexts are independent, with no semantic dependencies between them, the semantic dependencies within "*The sky is blue.*" should remain unchanged regardless of their surrounding context in the input sequence.

Specifically, given two input token sequences are $\mathbf{z}^{0(s_1)} = \{\mathbf{z}_i^0\}_{i=1}^{N_1}$ and $\mathbf{z}^{0(s_2)} = \{\mathbf{z}_j^0\}_{j=1}^{N_2}$, respectively. Here, we validate the semantic dependencies within $\mathbf{z}^{0(s_1)}$. We created two additional token sequences: $\mathbf{z}^{0(\text{Left})} = [\mathbf{z}^{0(s_2)}, \mathbf{z}^{0(s_1)}]$ and $\mathbf{z}^{0(\text{Right})} = [\mathbf{z}^{0(s_1)}, \mathbf{z}^{0(s_2)}]$, where $\mathbf{z}^{0(\text{Left})}$ is obtained by concatenating $\mathbf{z}^{0(s_2)}$ to the left and $\mathbf{z}^{0(\text{Right})}$ is obtained by concatenating $\mathbf{z}^{0(s_2)}$ to the right. For token $\mathbf{z}_i^0$ from $\mathbf{z}^{0(s_1)}$, we obtain the corresponding estimated semantic dependency token group $\hat{G}_{\mathbf{z}_i^0}^{s_1}$ via Eq. (6). By using the same approach, estimated semantic dependency token groups $\hat{G}_{\mathbf{z}_i^0}^{\text{Left}}$ and $\hat{G}_{\mathbf{z}_i^0}^{\text{Right}}$ for $\mathbf{z}^{0(\text{Left})}$ and $\mathbf{z}^{0(\text{Right})}$ can also be obtained. Then the Dependency Alteration Score (DAS) of $\hat{G}_{\mathbf{z}_i^0}^{\text{Left}}$ and $\hat{G}_{\mathbf{z}_i^0}^{s_1}$ can be calculated as follows:

$$\text{DAS}(\hat{G}_{\mathbf{z}_i^0}^{\text{Left}}, \hat{G}_{\mathbf{z}_i^0}^{s_1}) = 1 - \frac{\text{LCS}(\hat{G}_{\mathbf{z}_i^0}^{\text{Left}}, \hat{G}_{\mathbf{z}_i^0}^{s_1})}{L}, \tag{8}$$

where $\text{LCS}(\cdot)$ is the length of the longest common subsequence. In our case, it represents the longest sequence of tokens that appear in the same order in both contexts, despite irrelevant context or order changes. The score $\text{DAS}(\hat{G}_{\mathbf{z}_i^0}^{\text{Left}}, \hat{G}_{\mathbf{z}_i^0}^{s_1})$ measures how the semantic dependency changes when appending irrelevant context $\mathbf{z}^{0(s_2)}$ to the left of the original sequence $\mathbf{z}^{0(s_1)}$. Similar $\text{DAS}(\hat{G}_{\mathbf{z}_i^0}^{\text{Right}}, \hat{G}_{\mathbf{z}_i^0}^{s_1})$ can be obtained, which measures the changes of semantic dependency when appending irrelevant context $\mathbf{z}^{0(s_2)}$ to the right.

**Semantic Dependency Analysis with Irrelevant Context Order Change**   For irrelevant context order change, we observe whether the token dependency in sentence *"The sky is blue."* alters when inputting the sentence with irrelevant context order change, e.g., *"The sky is blue. The apple is red."* and input *"The apple is red. The sky is blue."*. We simply use $\text{DAS}(\hat{G}_{\mathbf{z}_i^0}^{\text{Left}}, \hat{G}_{\mathbf{z}_i^0}^{\text{Right}})$ to measure how the semantic dependency changes when appending the irrelevant context $\mathbf{z}^{0(s_2)}$ to the left and the right of the original sequence $\mathbf{z}^{0(s_1)}$.

**Experiment**   We conducted the semantic dependency analysis across over 5,000 cases to examine the impact of irrelevant context added to both the left and right sides, as well as the effect of sequence order changes, in order to determine whether semantic information propagation is context-dependent and order-dependent. Specifically, we measured the dependency changes when perturbing the token $\mathbf{z}_i^{0(s_1)}$ in the original sequence $\mathbf{z}^{0(s_1)}$. This involves evaluating the dependency alterations of its semantically dependent token groups by aligning the top 5 semantically dependent token groups ($L = 5$) and by aligning all tokens from the original sequence $\mathbf{z}^{0(s_1)}$ ($L = N_1$). The average dependency alteration scores are presented in Figure 2.

Figure 2(a) and Figure 2(b) illustrate the changes in semantic dependency when irrelevant context is appended on the left or right side. It shows that the rank of semantic dependency strength of common token is significantly affected by the context, while relationships of semantically more related tokens (Top 5) remain relatively stable.

Figure 2(c) further compares the changes in dependency when irrelevant context is added to the left versus the right side of the original sentence. The results reveal that adding context to the left side generally results in a greater alteration of semantic dependencies compared to the right side. This suggests that the order of irrelevant context can differentially impact the model's semantic dependency structures.

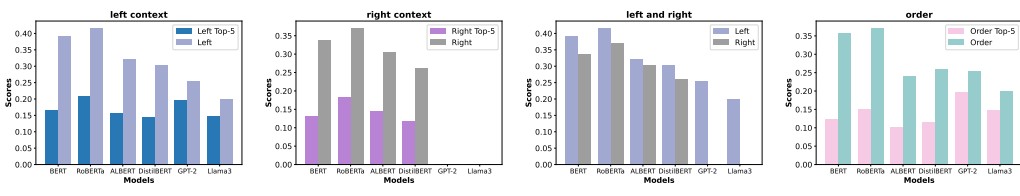

(a) Left context change     (b) Right context change     (c) Left & right context     (d) Order change

Figure 2: Semantic Dependency Alteration Score when irrelevant context or context order changes.

Figure 2(d) demonstrates the impact of altering the sequence order on semantic dependencies. The results also show that irrelevant token groups are easily influenced by unrelated contexts, while semantically more dependent tokens exhibit greater resilience to such alterations.

Overall, our findings indicate that both the introduction of irrelevant context and the modification of sequence order dramatically influence semantic information dependence within sentences. These results reinforce the importance of context placement and order in shaping the semantic dependency structures learned by Transformer-based language models

## 5   WHEN THE MODEL MAKES MISTAKES, IT FALSELY AGGREGATES SEMANTICALLY INDEPENDENT INFORMATION WITHIN A TOKEN

Transformer-based language models have demonstrated remarkable capabilities in various natural language tasks but occasionally produce incorrect answers. We hypothesize that such errors arise from the model's tendency to falsely aggregate dependent semantic information across tokens within transformer layers. Intuitively, in the final layer, the tokens are combined to produce the output probabilities via a linear prediction layer. However, the linear nature of this prediction layer limits its discriminative power, making it susceptible to errors when false dependencies are present. When a model erroneously aggregates semantic information from unrelated or misleading tokens, it can disproportionately influence the final token probabilities, leading to incorrect predictions. In this section, we try to verify our hypothesis.

**Evaluation of False Dependencies**   To test our hypothesis that model errors often result from falsely aggregated independent semantic information within tokens, we simply view model's wrong output token and question token as a false dependency for evaluation. Specifically, we compare the semantic dependencies between tokens in incorrect answers and question tokens against those in correct answers within a question-answering (QA) task. We analyze instances where the language model outputs either the correct answer extracted from the context or an incorrect one.

Consider the QA example illustrated in Figure 3, where the context provides the correct answer "national anthem" and an alternative phrase "sign language." If the BERT model incorrectly outputs "sign language" instead of "national anthem," this presents an opportunity to examine the underlying semantic dependencies that led to the error.

Formally, let $Q = \{\mathbf{q}_i^0\}_{i=1}^{N_Q}$ represent the set of tokens in the question, $A_{\text{correct}} = \{\mathbf{a}_i^0\}_{i=1}^{N_C}$ represent the correct answer tokens in the context, and $A_{\text{wrong}} = \{\mathbf{a}_i^0\}_{i=1}^{N_W}$ represent the incorrect answer tokens in the context. For each answer token $\mathbf{a}_i$, we measure its semantic information dependence on each question token $\mathbf{q}_j \in Q$ by computing a semantic dependence score $\Delta_{\mathbf{q}_j^L | \mathbf{a}_i^0}$. This score quantifies the degree to which answer token $\mathbf{a}_i$ influences the question token $\mathbf{q}_j$ in the final layer $L$ of the model. Next, we determine the maximum semantic dependence score for each answer token by selecting the highest $\Delta_{\mathbf{q}_j^L | \mathbf{a}_i^0}$ across all question tokens $\Delta'_{\mathbf{a}_i^0 | Q} = \max_{j=1}^{N_Q} \Delta_{\mathbf{q}_j^L | \mathbf{a}_i^0}$.

For both correct and incorrect answers, we compute the highest dependence scores across all respective answer tokens:

$$\Delta'_{A_{\text{correct}}|Q} = \max_{k=1}^{N_C} \Delta'_{\mathbf{a}_k^0}, \quad \Delta'_{A_{\text{wrong}}|Q} = \max_{k=1}^{N_W} \Delta'_{\mathbf{a}_k^0}.$$

To evaluate whether the maximum dependence score for incorrect answers exceeds that of correct answers when a model makes mistakes, we calculate the percentage that $\Delta'_{A_{\text{wrong}}|Q}$ is greater than

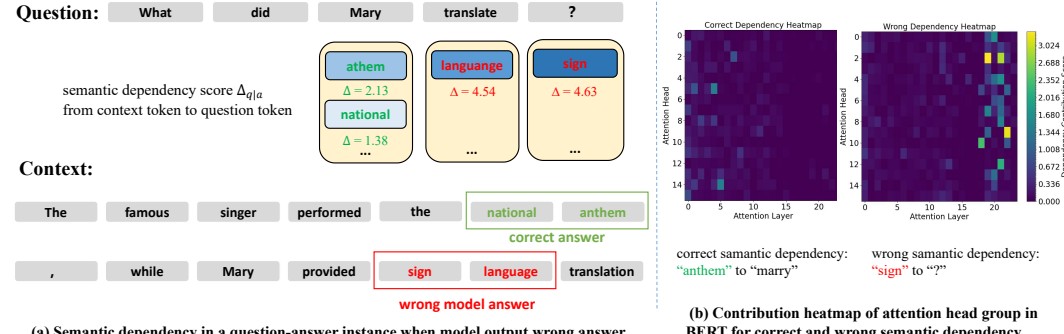

Figure 3: A question-answer instance for false semantically dependent information within tokens.

$\Delta'_{A_{\text{correct}|Q}}$ given the question and answer pairs where the model makes mistakes. Specifically,

$$P(f_\theta) = \sum_{i=1}^{H} \mathbb{1}_{\{\Delta'_{A_{\text{wrong}|Q}} > \Delta'_{A_{\text{correct}|Q}}\}},$$

where $H$ represents the total number of incorrect QA instances.

**Experiment** We apply our evaluation method to the Stanford Question Answering Dataset (SQuAD) 1.1 (Rajpurkar et al., 2016), which comprises context paragraphs extracted from Wikipedia articles, along with manually crafted questions and their corresponding correct answers. Each QA instance in the dataset provides a context from which the correct answer is a continuous span of text, which means the answer exists verbatim in the context. Our analysis involves processing over 100,000 QA validation cases across various Transformer-based models, including BERT, RoBERTa, ALBERT, Distil-BERT, Llama3, and GPT-2.

Table 3: Percentage of model output matching our information propagation assumption.

| Model | Percentage (%) |
|---|---|
| BERT | 79.07 |
| RoBERTa | 77.94 |
| ALBERT | 71.86 |
| DistilBERT | 81.87 |
| Llama3 | 64.56 |

For each QA instance, we first determine whether the model outputs an incorrect answer by evaluating the F1 score between the model's predicted answer and the ground truth answer. We consider a prediction to be incorrect if the F1 score is below 0.6. Consequently, we collect these incorrect answer cases (where F1 < 0.6) for further analysis to examine the presence of false dependencies. This selection criterion ensures that we focus on substantial errors rather than minor discrepancies, thereby providing a robust basis for evaluating semantic dependency misalignments.

In these selected cases, we identify the semantic dependencies between question tokens and both correct and incorrect answer tokens. For each incorrect answer token, we compute its semantic dependence score on question tokens and compare it with the dependence scores of correct answer tokens. Specifically, we calculate whether the maximum dependence score of incorrect answer tokens exceeds that of correct answer tokens. This comparison allows us to assess whether the model's errors are associated with falsely aggregated semantic dependencies from incorrect tokens influencing question tokens. The results are summarized in Table 3, which generally shows a significant proportion of model error cases across various models can be attributed to falsely aggregated semantic dependencies.

These findings demonstrate that a substantial majority of model errors are associated with stronger semantic dependencies from incorrect answer tokens compared to correct ones. For instance, in BERT's case, the high percentage implies that when the model selects an incorrect answer, it is more likely due to the erroneous answer tokens causing a greater semantic influence on the question tokens than the correct answer tokens. This misalignment in dependency strengths leads the model to favor incorrect information over the correct, contextually relevant answer.

The variation in probabilities across different models highlights inherent differences in how each architecture manages semantic dependencies and mitigates the impact of misleading information. Models like DistilBERT and BERT, with higher probabilities, may have architectural or training

advantages that make them more susceptible to false dependencies when errors occur. On the other hand, Llama3's lower percentage suggests a potentially more robust mechanism for distinguishing between relevant and irrelevant semantic information, thereby reducing the likelihood of false dependencies influencing its outputs.

**Localize Attention Head Group Responsible for Semantic Dependency**    Inspired by Gandelsman et al. (2024), the contribution of $l$-th MHA on $j$-th token can be broken down into tokens and heads.

$$\text{MHA}_j^l(\mathbf{Z}^{l-1}) = \sum_{h=1}^{H} \sum_{i=1}^{N} x_i^{l,h}, \quad x_i^{l,h} = \alpha_i^{l,h} W_{VO}^{l,h} z_i^{l-1} \tag{9}$$

Specifically, for any token dependency, i.e., token dependency from $i$-th token to $j$-th token, including correct or wrong token dependency in QA task mentioned above, we replace the $i$-th the token with $K$ randomly sampled tokens. Then we measure each head's contribution on semantic information dependency by calculating average change $\Delta_{\mathbf{z}_j^L|\mathbf{z}_i^0}^{l,h}$ between original head contribution and perturbed head contributions as follows:

$$\Delta_{\mathbf{z}_j^L|\mathbf{z}_i^0}^{l,h} = \frac{1}{K} \sum_{k=1}^{K} \left\| x_i^{l,h(k)} - x_i^{l,h(\text{org})} \right\|_2 \tag{10}$$

As is shown in figure 3(b), we test the dependency contribution score $\Delta_{\mathbf{q}_j^L|\mathbf{a}_i^0}^{l,h}$ of each attention head in BERT for both wrong semantic dependency between "sign"and "?" and correct semantic dependency between "anthem" to"marry" in corresponding QA instance in figure 3(a). In this case we can observe there are a group of attention heads (highlighted with bright color in the contribution heatmap) mutually contribute to the semantic dependency.

**Limitations and Future Work**    There are some limitations in our current method, which we believe present valuable opportunities for future work. Firstly, our analysis relies on perturbation-based approaches to assess token dependencies, which require that the answer tokens appear within the question. This constraint limits our ability to evaluate scenarios where the model generates answer tokens that are not directly present in the input question. We aim to expand our ability to effectively analyze dependencies in such cases to broaden the scope of our evaluations.

Additionally, perturbation inherently involves both the removal of existing information and the introduction of new information. The newly introduced information can lead to varying levels of variability in the output layer tokens. For example, if a token in the input sentence is replaced with a semantically similar but slightly different token, the model's response might vary significantly depending on how it interprets the new context. We mitigate this by employing random sampling of new tokens to ensure diversity and minimize bias; however, this approach may not fully eliminate all sources of variability. Future research will focus on refining this calibration. Thirdly, the influence of the last linear prediction layer can also affect our analysis. Although its discriminative power is limited due to its linear nature, some false dependencies in the last layer of tokens can be disentangled. As a result, certain false dependencies might be less influential on the final prediction. We believe that the score could be higher if the impact of the last layer on false dependencies is taken into account and would like to further explore this in future work.

## 6    CONCLUSION

In this paper, we delved into the internal mechanisms of transformer models to explore how semantic information is propagated and aggregated across tokens, which can contribute to the errors produced by large language models (LLMs). We show several key findings. Firstly, most tokens primarily retain their original semantic information throughout the layers of the transformer, indicating a strong connection to their initial meanings. Secondly, semantically dependent information tends to be encoded together within a token, reflecting the model's ability to capture related concepts. Thirdly, we observed that the aggregation of semantic information is influenced by both irrelevant context changes and the order of token sequences, highlighting potential areas for model refinement. Lastly, our findings revealed that when models make mistakes, tokens encode incorrect semantic dependency. We believe these insights offer valuable implications for future transformer model design.

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

# A APPENDIX

## A.1 RELATED WORKS

**Semantic Information Flow in Transformer** Existing work (Liao et al., 2021; Schuster et al., 2022; Elhoushi et al., 2024) have studied model activation stability in later layers of transformer models. Specifically, additional layers may contribute minimally to the refinement of token representations, which enables techniques like early exit to accelerate inference. However, whether token in the last layer contains its original semantic information in the input layer has not been studied. Geva et al. (2023) analyze how factual associations are recalled in auto-regressive language models, highlighting the roles of MLP sublayers in enriching subject representations and attention heads in extracting attributes. While our study address a gap by studying how semantic information flow between tokens through attention layers in both non-auto-regressive (BERT) and auto-regressive models (GPT, Llama).

**Irrelevant and Adversarial Context Influence** Robustness studies have demonstrated that the inclusion of irrelevant context (Shi et al., 2023) or adversarial sentences (Jia & Liang, 2017) in prompts can lead to a significant decline in model accuracy. They usually works by by analyzing model performance on various types of adversarial examples and attribute the decline to broader issues, such as the model's tendency to rely on surface-level features like word overlap and positional cues. Our study provide an underlying reason for such performance decline from a token-level perspective. Specifically, We found the rank of different semantic dependency strength encoded in a token changes when adding irrelevant context or simply change the order of the context sequence. Our insight can further help training or finetuning a robust language model in which the rank of encoded semantic dependency within tokens is stable when given irrelevant or adversarial context in prompts.

**Interpretable Model Error Based on Attention Heads** Existing works have studied specific roles of attention head to explain model errors. Wu et al. (2024) identifies specific attention heads, termed retrieval heads, which are critical for retrieving factual information from long contexts. The absence or malfunctioning of these retrieval heads may lead to model errors. Gandelsman et al. (2024) shows some attention heads in CLIP have property-specific roles (e.g., location or shape), which are important for model performance. Our study addresses another reason by exploring how token-level semantic dependency influences model mistakes, which provide another critical perspective on understanding and correcting model mistakes under specific question answering cases.

**Probing Study for Linguistic Properties in Transformer** Probing methods (Rogers et al., 2021) are widely used to analyze the internal representations of pre-trained language models to determine whether specific linguistic properties are encoded. Hewitt & Manning (2019) demonstrated that BERT encodes syntactic tree structures in its vector space, allowing a probing classifier to reconstruct syntactic distances between words using linear transformations. Tenney (2019) revealed that BERT encodes high-level linguistic features like entity types, semantic roles, and relations through probing tasks. Pimentel et al. (2020) utilized information-theoretic probing methods to quantify the mutual information between model representations and linguistic properties, reducing over-interpretation risks. Wu et al. (2020) proposed a parameter-free probing technique that analyzed the influence of syntactic subtree structures on MLM predictions.

These works primarily investigate how models encode syntactic and high-level semantic features, such as entity relations or syntactic structures. In contrast, our study focuses specifically on token-level semantic dependencies, analyzing fine-grained interactions between individual tokens rather than task-specific feature aggregation or high-level semantic encoding. Moreover, we introduce an evaluation framework to measure semantic dependency strength between two tokens without relying on prior knowledge. Our approach also identifies false semantic dependencies that arise when the model produces incorrect answers. Unlike static syntactic or semantic structures, our framework captures the dynamic and context-sensitive semantic dependencies, which can vary irregularly across diverse scenarios.

Table 4: Percentage of a token propagates semantic information to other tokens.

| | gsm8k | Yelp | GLUE | DailyMail | OpenOrca | WikiText |
|---|---|---|---|---|---|---|
| BERT | 99.44 | 99.09 | 99.16 | 99.20 | 99.34 | 99.52 |
| RoBERTa | 96.46 | 96.42 | 97.04 | 96.42 | 95.98 | 96.07 |
| ALBERT | 97.88 | 97.99 | 98.35 | 97.34 | 98.23 | 96.63 |
| DistilBERT | 95.44 | 95.89 | 96.37 | 96.29 | 96.42 | 95.93 |
| GPT-2 | 100.00 | 100.00 | 100.00 | 100.00 | 100.00 | 100.00 |
| LLama3 | 100.00 | 100.00 | 100.00 | 100.00 | 100.00 | 100.00 |

## A.2 EXPERIMENT DETAILS

**Percentage of a token propagates semantic information to other tokens.** We also observe the change of the specific input word causes influence on other token outputs in the final layer in experiment of Section 2. The result over all cases (each token perturbation is treated as one case, over 600,000 cases are evaluated for each model) is shown in Table 4. Even if minor, in models like BERT, the change is almost 100%, which means each token receives pieces of semantic information from other tokens. While in auto-regressive models like Llama or GPT, the token only influences the tokens on this token's right side. We observe the changes of tokens on each tokens' left side is 0. we can also observe the change exists in all tokens on each token's right side, which suggests each token receives pieces of semantic information from tokens on its left side.

**Why Using Neural Dependency Parsing Tool in Section 3** Noted that our analysis relies on semantic dependency data derived with SpaCy, a pretrained neural network-based dependency parser. SpaCy generates syntactic dependency trees using robust neural architectures trained on large annotated corpora, offering a reliable approximation of semantic dependencies. To our knowledge, no token-level semantic dependency dataset with comprehensive human annotations exists. Constructing such a dataset would be prohibitively expensive and prone to omissions due to the complexity of identifying all dependent token relationships manually. Thus, we use neural dependency parsing tool to generate a specialized semantic dependency datasets for our experiment.

**Why Using Longest Common Subsequence in Section 4** Consider a simple example to understand how LCS captures changes in token order: Suppose we have two sequences, $A = [1, 2, 3, 4]$ and $B = [2, 3, 4, 1]$. In moving from sequence $A$ to sequence $B$, the order of the tokens changes such that the token "1" moves from the beginning to the end. Here, the LCS between $A$ and $B$ is the subsequence [2, 3, 4], which has a length of 3. This subsequence represents the largest set of tokens that have retained their original order between the two sequences. Since the total number of tokens, $N$, is 4, the LCS length of 3 indicates that one token ("1") changed its position relative to the others. By calculating DAS $= 0.25$, we find that a quarter of the token order has been altered due to the change in context. Thus, a lower LCS value (relative to $N$) results in a higher DAS, reflecting a more significant change in token dependency patterns. This metric effectively highlights how sensitive the token dependencies are to contextual modifications, demonstrating the dynamic nature of semantic processing in natural language systems.

**Discussion on Experiment Results in Section 5** The result in Table 3 shows a significant proportion of model error cases across various models can be attributed to falsely aggregated semantic dependencies in general. Specifically, BERT exhibits a percentage of 79.07%, indicating that in approximately 79% of its incorrect answer cases, the semantic dependencies from incorrect answer tokens to question tokens surpass those from correct answer tokens. This suggests that when BERT makes an error, it is predominantly influenced by misleading semantic information from incorrect tokens. Similarly, RoBERTa and ALBERT show probabilities of 77.94% and 71.86%, respectively, reinforcing the trend that false dependencies significantly contribute to model errors across different Transformer architectures. DistilBERT stands out with the highest percentage of 81.87%, suggesting an even greater tendency for incorrect dependencies to influence its erroneous answers. Conversely, the autoregressive model Llama3 exhibits the lowest percentage at 64.56%, indicating a relatively lower incidence of false dependency aggregation in its incorrect outputs. It leaves an area for further exploration to understand the underlying mechanism responsible for this performance.

## A.3 PESUDOCODE FOR SECTION 5

---

**Algorithm 1** Evaluation of False Dependencies

---

**Data:** dataset with $M$ instances, Transformer model $f_\theta$, number of perturbations $K$
**Result:** Percentage $p$ that perturbing the $i$-th token predominantly affects its own output token
Initialize $count\_correct \leftarrow 0$;

**for** *each incorrect QA instance* $m = 1$ *to* $H$ **do**

    Extract question tokens $Q = \{\mathbf{q}_i^0\}_{i=1}^{N_Q}$, correct answer tokens $A_{\text{correct}} = \{\mathbf{a}_i^0\}_{i=1}^{N_C}$, and incorrect answer tokens $A_{\text{wrong}} = \{\mathbf{a}_i^0\}_{i=1}^{N_W}$;

    **for** *each answer token* $\mathbf{a}_k^0 \in A_{correct} \cup A_{wrong}$ **do**

        **for** $k = 1$ *to* $K$ **do**

            **if** $k = 1$ **then**

                $\tilde{\mathbf{z}}_k^0 \leftarrow \mathbf{a}_k^0$ ;                          // Original token

            **else**

                $\tilde{\mathbf{z}}_k^0 \leftarrow \text{RandomToken}(\mathcal{V})$ ;             // Perturbed token

            **end**

            Construct perturbed sequence $\tilde{\mathbf{z}}^{0(k)}$ by replacing $\mathbf{a}_k^0$ with $\tilde{\mathbf{z}}_k^0$;

            Compute final layer representations $\tilde{\mathbf{z}}^{L(k)} \leftarrow f_\theta(\tilde{\mathbf{z}}^{0(k)})$;

        **end**

        Compute original final layer representations $\mathbf{z}^{L(\text{org})} \leftarrow f_\theta(\mathbf{z}^{0(\text{org})})$;

        **for** *each token* $j = 1$ *to* $N$ **do**

            Calculate $\Delta_{\mathbf{z}_j^L|\mathbf{a}_k^0} \leftarrow \frac{1}{K-1} \sum_{k=2}^{K} \left\| \tilde{\mathbf{z}}_j^{L(k)} - \mathbf{z}_j^{L(\text{org})} \right\|_2$;

        **end**

        Determine maximum dependency score for $\mathbf{a}_k^0$:

        $\Delta'_{\mathbf{a}_k^0|Q} = \max_{j=1}^{N_Q} \Delta_{\mathbf{q}_j^L|\mathbf{a}_k^0}$

    **end**

    Determine maximum dependency score for correct answers:

$$\Delta'_{A_{\text{correct}}|Q} = \max_{k=1}^{N_C} \Delta'_{\mathbf{a}_k^0}$$

    Determine maximum dependency score for wrong answers:

$$\Delta'_{A_{\text{wrong}}|Q} = \max_{k=1}^{N_W} \Delta'_{\mathbf{a}_k^0}$$

    **if** $\Delta'_{A_{wrong}} > \Delta'_{A_{correct}}$ **then**

        $count\_correct \leftarrow count\_correct + 1$;

    **end**

**end**
Calculate percentage:

$$p(f_\theta) = \frac{count\_correct}{M}$$

**return** $p(f_\theta)$;

---

## A.4 LOCALIZE SEMANTIC DEPENDENCY WITHIN ATTENTION LAYERS IN SECTION 5

To further explore how network contributes to model errors, we have developed a method to identify the attention heads primarily responsible for specific token dependency. Here, we present the intuition and detailed equations of how we localize semantic dependency within attention layers.

Intuitively, when input token carrying specific semantic information changes, the attention heads relevant to corresponding semantic information propagation will exhibit significant changes in their outputs, while the outputs of irrelevant heads will remain relatively unchanged. Therefore, by identifying heads with the highest variation in their contribution on given token dependency, we can pinpoint the group of attention heads that are mutually responsible for any token dependency including wrong or correct token dependency in QA task.

As mentioned in Eq. (2), transformer encoder or transformer decoder is a residual network built from $L$ layers, each of which contains a multi-head self-attention (MHA) followed by feed forward network (FFN) block.

In $l$-th MHA layer, the input stream $z^{l-1}$ is processed separately by $H$ attention heads. Specifically, the input sequence $Z^{l-1}$ is separately projected into $Q$, $K$, $V$ matrix in $h$-th attention head of $l$-th layer as follows:

$$\mathbf{Q}^{l,h} = \mathbf{Z}^{l-1}\mathbf{W}_Q^{l,h}, \quad \mathbf{K}^{l,h} = \mathbf{Z}^{l-1}\mathbf{W}_K^{l,h}, \quad \mathbf{V}^{l,h} = \mathbf{Z}^{l-1}\mathbf{W}_V^{l,h} \tag{11}$$

Then attention weight matrix $\mathbf{A}^{l,h} \in \mathbb{R}^{N \times N}$ is calculated as follows:

$$\mathbf{A}^{l,h} = \text{softmax}\left(\frac{\mathbf{Q}\mathbf{K}^T}{\sqrt{d_k}}\right) \tag{12}$$

The output of each attention head is

$$\mathbf{O}^{l,h} = \mathbf{A}^{l,h}\mathbf{V}^{l,h} \tag{13}$$

For multi-head attention, the outputs of each head are concated and projected to $Z^l \in \mathbb{R}^{N \times D}$, where $W_O$ is output weight matrix.

$$\mathbf{MHA}^l(\mathbf{z}^{l-1}) = \text{Concat}(\mathbf{O}^{l,1}, \mathbf{O}^{l,2}, \ldots, \mathbf{O}^{l,H})\mathbf{W}_O \tag{14}$$

The class token and the other tokens share the same computation process. Inspired by Gandelsman et al. (2024), the contribution of $l$-th MHA on $j$-th token can be broken down into tokens and heads. We can observe that given a token, each context token contribute to this token by adding operation for semantic information aggregation, which generate context related token representation.

$$\mathrm{MHA}_j^l(\mathbf{Z}^{l-1}) = \sum_{h=1}^{H}\sum_{i=1}^{N} x_i^{l,h}, \quad x_i^{l,h} = \alpha_i^{l,h} W_{VO}^{l,h} z_i^{l-1} \tag{15}$$

specifically, for any token dependency, i.e., token dependency from $i$-th token to $j$-th token, including correct or wrong token dependency in QA task mentioned above, we replace the $i$-th the token with $K$ randomly sampled tokens. Then we measure each head's contribution on semantic information dependency by calculating average change $\Delta_{\mathbf{z}_j^L|\mathbf{z}_i^0}^{l,h}$ between original head contribution and perturbed head contributions as follows:

$$\Delta_{\mathbf{z}_j^L|\mathbf{z}_i^0}^{l,h} = \frac{1}{K}\sum_{k=1}^{K}\left\|x_i^{l,h(k)} - x_i^{l,h(\text{org})}\right\|_2 \tag{16}$$

As is shown in figure 3(b), we test the dependency contribution score $\Delta_{\mathbf{q}_j^L|\mathbf{a}_i^0}^{l,h}$ of each attention head in BERT for both wrong semantic dependency between "sign"and "?" and correct semantic dependency between "anthem" to"marry" in corresponding QA instance. In this case we can observe there are a group of attention heads (highlighted with bright color in the contribution heatmap) mutually contribute to the semantic dependency. We can also find the head group responsible for wrong dependency is clearly more bright than correct dependency, showing a different pattern.

**Discussion**   In our experiment, we found the model's attention head performance for semantic information storage is different in various QA cases, thus unable to unify a group of specific heads for general model mistakes. We will further explore the general pattern in the future. Additionally, Geva et al. (2023) have shown MLPs also encode enriched representations that propagate attributes. Such representations may inadvertently amplify irrelevant or erroneous semantic information. We aim to extend our analysis to quantify the contribution of MLPs to semantic dependency in the future.

## A.5 SYMBOL LIST

Table 5: Symbol List and Their Explanations

| Symbol | Explanation |
|---|---|
| $\mathbf{z}_i^l$ | The embedding of the $i$-th token in the $l$-th layer. |
| $\mathbf{z}_j^l$ | The embedding of the $j$-th token in the $l$-th layer. |
| $\mathbf{z}_i^{l(\text{org})}$ | The original embedding of the $i$-th token in the $l$-th layer. |
| $\tilde{\mathbf{z}}_i^{L(k)}$ | The $k$-th perturbed embedding of the $i$-th token in the $l$-th layer. |
| $\Delta_{\mathbf{z}_j^L|\mathbf{z}_i^0}$ | Semantic dependency score, which measures how the perturbation of token $i$ at layer 0 affects token $j$ at the final layer $L$. |
| $N$ | The number of tokens in a token sequence. |
| $K$ | The $i$-th token in layer 0 is perturbed $K$ times to calculate average change of the i-th token in layer $L$. $K = 5$ in our experiments. |
| $M$ | The number of total perturbed token cases across all sequences we evaluate. |
| $P(f_\theta)$ | Percentage $P$ of the cases that the transformer-based language model $f_\theta$ matches our finding. |
| $W_{\mathbf{w}_i^0}$ | True semantically dependent word group for the $i$-th word in layer 0 based on semantic dependency parsing. |
| $G_{\mathbf{z}_i^0}$ | Truthful semantically dependent token group for the $i$-th token in layer 0 based on semantic dependency parsing. |
| $\hat{G}_{\mathbf{z}_i^0}$ | Estimated semantically dependent token group for the $i$-th token using token perturbation. |
| $K_{\text{top}}$ | The number of top tokens most sensitive to the perturbation of the input token. $K_{\text{top}}$ is set to the size of $G_{\mathbf{z}_i^0}$. In the experiment, we evaluate the overlap of $G_{\mathbf{z}_i^0}$ and top 5 tokens when the size are under 5. |
| $S_{z_i^0}$ | Alignment score between the truthful ($G_{z_i^0}$) and estimated ($\hat{G}_{z_i^0}$) semantically dependent token groups. |
| $\hat{G}_{\mathbf{z}_i^0}^{s_1}, \hat{G}_{\mathbf{z}_i^0}^{s_2}$ | Estimated semantically dependent token group for the $i$-th token corresponding to token sequences $s_1$ and $s_2$. |
| $\hat{G}_{\mathbf{z}_i^0}^{\text{Left}}, \hat{G}_{\mathbf{z}_i^0}^{\text{Right}}$ | Estimated semantically dependent token group for the $i$-th token corresponding to concatenated sequences $(s_2, s_1)$ and $(s_1, s_2)$. |
| $\text{DAS}(\cdot)$ | Dependency Alteration Score, measuring the impact of irrelevant context or sequence order changes on semantic dependencies in a sequence. |
| $L$ | The number of chosen semantically dependent tokens in the original token sequence $\mathbf{z}^{0(s_1)}$. e.g., $L = 5$ when choosing top 5 semantically dependent tokens for evaluation. |
| $\mathbf{q}_i^l$ | The embedding of the $i$-th question token in the $l$-th layer. |
| $\mathbf{a}_i^l$ | The embedding of the $j$-th answer token in the $l$-th layer. |
| $\Delta_{\mathbf{q}_j^L|\mathbf{a}_i^0}$ | Semantic dependency score in QA task, which measures how the perturbation of $i$-th answer token at layer 0 affects $j$-th question token at the final layer $L$. |
| $\Delta'_{a_i^0|Q}$ | Highest semantic dependence score above all semantic dependency between all question tokens and $i$-th answer tokens in a QA task. |
| $\Delta'_{A_{\text{correct}|Q}}, \Delta'_{A_{\text{wrong}|Q}}$ | Highest semantic dependence score above all semantic dependency between question tokens and answer tokens (correct or wrong) in a QA task. |

