# OpenReview forum: "Understanding Mistakes in Transformers through Token-level Semantic Dependencies"
_ICLR.cc/2025/Conference — Submitted to ICLR 2025_

### Official Review · Reviewer_mBPy · 2024-11-01

**Soundness:** 2
**Presentation:** 2
**Contribution:** 2
**Rating:** 3
**Confidence:** 3

**Summary:**

This paper uses perturbation analysis to understand how individual tokens in the input of a Transformer affect the contextualized representations at the output of a Transformer. Essentially, they randomly replace various input tokens and measure the L2 difference on the encoded token representations. The paper studies how these dependencies relate to linguistic notions of semantic dependency. The paper also studies how seemingly irrelevant context affects the contextualized representations. Finally, the paper studies the correlation in prediction errors with deviations from the expected dependency relations.

**Strengths:**

* It was nice to see comparisons across models spanning from BERT to LLama3. It was interesting to see trends across the various analyses with respect to modeling advancements across time.

* It’s potentially interesting to connect errors in model predictions to errors in propagating information correctly between tokens.

**Weaknesses:**

Overall weaknesses:

* The paper is structured as a sequence of loosely related experiments and analysis. It was a bit difficult to understand which findings were new and interesting. I think the paper could be improved by *removing* some content or deferring it to the appendix in order to focus and expand on the most interesting results.
* It would have been nice to contextualize the methods and findings more in prior work, e.g. probing studies (e.g. summarized in Rogers et al. 2020, https://arxiv.org/abs/2002.12327) and studies related to robustness to adversarial changes in context (e.g. Jia & Liang 2017, https://arxiv.org/abs/1707.07328). It would have been nice to have included discussion of in what cases the authors' findings are validating those of prior work vs. contradicting.
* Additionally, in many cases it seems the authors invent new metrics or experimental methodologies without discussing how such metrics or methods compare with those proposed by prior work.

Section 2

* The definition of `P(f_\theta)` was difficult to parse. Should `M` be `N` and `m` be `i`? It might also be helpful to remind the reader of the indicator function notation being used.
* It seems like it would make more sense to refer to this quantity as a “percentage” than a “probability”? (Here and elsewhere in the paper, too)
* Are there metrics from prior work that could have been chosen instead to measure the saliency of various input tokens on contextualized representations in the output?
* It wasn’t clear what the take-away was for the finding here. Should we be surprised that the most salient input for a given output token is the corresponding input token?
* Given the significance of casual attention mask on the analysis, it might be worth explicitly denoting in Tables 1 and 2 which models are decoder-only vs. encoder-only.

Section 3

* This analysis relies on a neural dependency parser, so although the analysis is presented as analyzing the degree to which dependencies in contextualized representations encode “semantic dependencies”, it might be helpful to clarify that the analysis is actual measuring agreement with a neural dependency parser model rather than some "ground truth" notion.
* It would have been useful to connect the methodology and findings with prior work that has studied the degree to which Transformer representations capture linguistic notions of syntactic or semantic dependencies.
* Prior work on probing could provide an alternative to the perturbation-based analysis, which has some limitations (as discussed by the authors towards the end of the paper).

Section 4

* I think this section is potentially the most interesting, because it could be interesting to attribute prediction errors to cases where the model appears to misunderstand the syntactic or semantic dependencies between tokens. However, I have some concerns about the methodology.
* It would be useful to show the % of cases where the incorrect answer has higher saliency than the correct answer for “correct cases” as a comparison point for Table 4.
* The experimental methodology seems a bit circular. The intended conclusion is that for cases where the model chooses the incorrect answer, the incorrect answer has higher saliency towards the question tokens than the correct answer. This doesn’t necessarily seem to be surprising, and it’s not clear what is actionable about this finding.

Nits

* The terms “unfaithful” and “semantically dependent information” appear in the abstract without clear definitions. It wasn’t clear to me what their meaning was in this context. Perhaps this could be clarified to improve readability of the abstract.

**Questions:**

Please see weaknesses.

---

> ### Author Response · Authors · 2024-11-25
>
> **1. Overall weaknesses**
>
> **Q1.1 [clarify presentation] The paper is structured as a sequence of loosely related experiments and analysis. It was a bit difficult to understand which findings were new and interesting. I think the paper could be improved by *removing* some content or deferring it to the appendix in order to focus and expand on the most interesting results.**
>
> Thank you for this suggestion. The four findings are progressively related. In the revision, we refine the importance of the four findings to make them more clear and add a brief introduction to their relationships in the Introduction. We also focus on the most impactful results by moving some trivial parts into the appendix.
>
> A1.1.1 [**clarify finding 1 importance:** Most tokens primarily retain their original semantic information, even as they pass through multiple layers.] Our finding focuses on whether the **$i$-th token of the last layer L** mostly contains its original semantic information from the **$i$-th token in Layer 0**. This has not been studied by existing works. This is a fundamental prerequisite for studying token-level semantic dependencies. If the semantic information of the $i$-th token in layer 0 is very different from the $i$-th token in the last layer, our later study on token semantic dependencies would be invalid. These results directly support the feasibility of studying how semantic information propagates between tokens in transformer architectures.
>
> A1.1.2[**clarify finding 2 importance**: Semantically dependent information is usually encoded together within a token.] Here, we focus on investigating **truthful semantic dependency** in tokens. The experiment for the second finding validates that tokens are primarily influenced by semantically dependent tokens, showing our perturbation-based method captures semantic dependencies, reinforcing the robustness and reliability of our approach.
>
> A1.1.3 [**clarify finding 3 importance**: The semantic dependency within a token is sensitive to even irrelevant changes in context and order of prompts.] The third finding reveals how the model’s performance is affected by irrelevant context at the token level, evidenced by changes in **the rank of tokens’ semantic dependencies**. Furthermore, we observed that even altering the position of irrelevant context can impact token relationships within the sequence while preserving the overall semantic meaning. This highlights how token-level analysis uncovers additional insights beyond performance metrics, providing alternative measures for evaluating model behavior.
>
> The above three findings are all prerequisites of studying how token-level semantic dependency influences model mistakes, which provide a critical perspective on understanding and correcting model mistakes.
>
> **Q1.2 [prior work]  Contextualizing findings in prior work: It would have been nice to contextualize the methods and findings more in prior work, e.g. probing studies (e.g. summarized in Rogers et al. 2020) and studies related to robustness to adversarial changes in context (e.g. Jia & Liang 2017). It would have been nice to have included discussion of in what cases the authors' findings are validating those of prior work vs. contradicting.**
>
> Thank you for this suggestion. We will enhance the discussion in related works by linking our findings to prior studies, such as probing (e.g., Rogers et al., 2020) and robustness studies (e.g., Jia & Liang, 2017).
>
> **A1.2.1 Probing Study**
>
> (Existing Work) Probing tasks (Rogers et al., 2020). have revealed how BERT encodes syntactic and semantic features across its layers, showing that lower layers focus on syntactic properties, while higher layers capture semantic dependencies. These studies often use lightweight classifiers or attention pattern analysis to uncover linguistic structures within BERT’s representations.
>
> (Difference with Our Work) However, probing tasks typically analyze static, pre-trained representations. In contrast, our work dynamically tests token-level semantic dependencies by introducing perturbations and measuring their effects on token semantic dependencies. This approach enables us to identify how falsely encoded semantic dependency lead to model mistakes. Moreover, we extend the scope of probing studies by examining the influence of irrelevant context specifically on the rank of token semantic dependenciy strength.

---

> ### Author Response · Authors · 2024-11-25
>
> **A1.2.2 Robustness Study**
>
> (Existing Work) Robustness studies (Jia & Liang, 2017) show that adversarial sentences can drastically reduce the performance of machine reading comprehension models. By analyzing model performance on various types of adversarial examples, they attribute the decline to broader issues, such as the model's tendency to rely on surface-level features like word overlap and positional cues.
>
> (Difference with Our Work) Our study seeks to explain this performance decline from the perspective of the model's internal mechanisms, providing an underlying reason for the observed accuracy drops in robustness studies. We found **the rank of semantic dependency strength encoded in a token changes** when adding irrelevant context or simply changing the order of the context sequence. We also provide a statistical evaluation method rather than observing the model’s output performance in a general study.
>
> (Importance)
> Our insight can further help training or finetuning a robust language model in which the rank of encoded semantic dependency within tokens is stable when given irrelevant context in prompts.
>
> **Q1.3 [prior work] Additionally, in many cases it seems the authors invent new metrics or experimental methodologies without discussing how such metrics or methods compare with those proposed by prior work.**
>
> Thank you for the suggestion. We will further discuss how our proposed metrics differ from or complement existing metrics in related works. We added related works and discussions in the appendix.
>
> **Section 2**
>
> **Q2.1 The definition of `P(f_\theta)` was difficult to parse. Should `M` be `N` and `m` be `i`? It might also be helpful to remind the reader of the indicator function notation being used.**
>
> We appreciate the suggestion regarding the notation. To clarify, in our definition of $P(f_\theta)$, M represents the number of tested token cases across all sequences, not the number of tokens N in a single sequence. Each test case may include multiple sequences, making M the broader measure. In the revision, we will add an explicit explanation of M to distinguish it from N for better reader comprehension.
>
> **Q2.2 It seems like it would make more sense to refer to this quantity as a “percentage” than a “probability”? (Here and elsewhere in the paper, too)**
>
> Thanks for your suggestion. we will change this in the later version.
>
> **Q2.3 Are there metrics from prior work that could have been chosen instead to measure the saliency of various input tokens on contextualized representations in the output?**
> Thank you for your question.
>
> A2.3.1 Saliency refers to the importance of specific input features (such as tokens) in contributing to the model’s final output. Semantic dependency is the relationship between words in a sentence where the meaning of one word depends on another word in the sentence
>
> A2.3.2 Our study mainly focuses on how **token-level semantic dependency** influences model performance through the token perturbation method,  which is different from works that study input tokens’ saliency on representation output.

---

> ### Author Response · Authors · 2024-11-25
>
> **Q2.4 [clarify our finding 1]It wasn’t clear what the take-away was for the finding here. Should we be surprised that the most salient input for a given output token is the corresponding input token?**
>
> Thank you for your concern. It is **not intuitive** that the most salient input token for a given output token is often the corresponding input token after extensive processing by deep transformer models.
>
> A2.4.1 (**Semantic Preservation Across Layers**) Our focus is studying how semantic information is preserved in tokens. Even after 24 (BERT-large) or 32 (Llama3-7B) attention layers most tokens still primarily retain their original semantic information. **This is surprising because**, during the attention mechanism, tokens aggregate diverse semantic information from other tokens in the sequence. The fact that most tokens still predominantly reflect their initial semantics highlights a strong retention property, which is not inherently expected given the iterative aggregation of contextual information.
>
> A2.4.2 **(Model-Specific Insights)** The observation that 20% of tokens in GPT-2 do not primarily retain their original semantic information points to **a potentially distinct mechanism** in how GPT-2 handles token-level information. This could indicate a different trade-off in how semantic context is integrated across tokens, which might relate to its performance characteristics compared to other models like BERT or LLaMA. Understanding this divergence could reveal alternative modeling strategies that influence downstream tasks differently.
>
> A2.4.3  (Importance) The finding that the last-layer token retains its original semantic information is critical for multiple reasons. It ensures that semantic dependency analyses in later layers are meaningful and reliable. It also provides a stable basis for interpretability, enabling attention-based methods to trace token contributions accurately. Additionally, this property could inspire new architectures that further optimize **semantic stability**.
>
> **Q2.5 Given the significance of casual attention mask on the analysis, it might be worth explicitly denoting in Tables 1 and 2 which models are decoder-only vs. encoder-only.**
>
> Thank you for this suggestion. We will explicitly denote which models are encoder-only (Bert series) and decoder-only (GPT, Llama).
>
> **Section 3**
>
> **Q3.1 [dependency parsing tool] This analysis relies on a neural dependency parser, so although the analysis is presented as analyzing the degree to which dependencies in contextualized representations encode “semantic dependencies”, it might be helpful to clarify that the analysis is actual measuring agreement with a neural dependency parser model rather than some "ground truth" notion.**
>
> Thank you for your suggestion.
>
> A3.1.1 (Dependency Parsing Tool) We have mentioned in our paper that this analysis relies on semantic dependency data derived from SpaCy, a pre-trained neural network-based dependency parser. SpaCy generates syntactic dependency trees using robust neural architectures trained on large annotated corpora, offering a reliable approximation of semantic dependencies.
>
> A3.1.2(the Benefit of Using This Tool) To our knowledge, no token-level semantic dependency dataset with comprehensive human annotations exists. Constructing such a dataset would be prohibitively expensive and prone to omissions due to the complexity of identifying all dependent token relationships manually. We will emphasize that our methodology measures the alignment of model-derived token dependencies with those provided by SpaCy as a proxy for ground truth.

---

> ### Author Response · Authors · 2024-11-25
>
> **Q3.2 [Connection with prior work] It would have been useful to connect the methodology and findings with prior work that has studied the degree to which Transformer representations capture linguistic notions of syntactic or semantic dependencies.**
>
> (our change in revision) Thank you for your suggestion. We will add related work in the later version as follows:
>
> A3.2.1 (existing work) There are several prior works investigating how Transformer-based models capture linguistic properties. Hewitt and Manning (2019) demonstrated that BERT encodes syntactic tree structures in its vector space, allowing a probing classifier to reconstruct syntactic distances between words. Tenney et al. (2019) revealed that BERT encodes high-level linguistic features like entity types, semantic roles, and relations through probing tasks. Pimentel et al. (2020) utilized information-theoretic probing methods to quantify the mutual information between model representations and linguistic properties, reducing over-interpretation risks. Wu et al. (2020) proposed a parameter-free probing technique that analyzed the influence of syntactic subtree structures on MLM predictions.
>
> A3.2.2 (Difference) These works primarily investigate how models encode **syntactic and high-level semantic features**, such as entity relations or syntactic structures. In contrast, our study focuses specifically on **token-level semantic dependencies**, analyzing fine-grained interactions between individual tokens rather than task-specific feature aggregation or high-level semantic encoding. Moreover, we introduce an evaluation framework to **measure semantic dependency strength** between two tokens without relying on prior knowledge. Our approach also identifies false semantic dependencies that arise when the model produces incorrect answers. Unlike static syntactic or semantic structures, our framework captures the dynamic and context-sensitive semantic dependencies, which can vary irregularly across diverse scenarios.
>
> Reference:
>
> [1] Hewitt, J., & Manning, C. D. 2019. A Structural Probe for Finding Syntax in Word Representations
>
> [2] Tenney et al. 2019. BERT Rediscovers the Classical NLP Pipeline
>
> [3] Wu et al. 2020. Perturbed Masking: Parameter-Free Probing of Linguistic Structure in MLMs
>
> [4] Pimentel et al. 2020. Information-Theoretic Probing for Linguistic Structure
>
> **Q3.3 [Alternative methodologies] Prior work on probing could provide an alternative to the perturbation-based analysis, which has some limitations (as discussed by the authors towards the end of the paper).**
>
> A3.3.1 We appreciate your suggestion. Our study mainly focuses on how token-level semantic dependency influences model performance through a token perturbation method, specifically how semantic information propagates from input tokens to last-layer tokens. This requires:
>
> - Analyzing **token-level interactions** rather than an aggregation of task-specific features or  high-level semantic features in models
> - Evaluating **semantic dependency strength** between two tokens without prior knowledge.
> - Evaluating false semantic dependency when the model makes mistakes
>
> A3.3.2 Probing methods cannot fully address these needs because they typically focus on **predefined linguistic tasks** (e.g., syntactic tree reconstruction or semantic role labeling) rather than dynamic semantic dependency shifts, which can not be used when semantic dependency are confusing and irregular, especially when we analyze why model make mistakes.
>
> **Section 4**
>
> **Q4.1 It would be useful to show the % of cases where the incorrect answer has higher saliency than the correct answer for “correct cases” as a comparison point for Table 4.**
>
> Thank you for your suggestion. However, the correct case is not provided with the wrong model answer as a reference. Thus it is impossible to get cases with incorrect answer saliency when the model outputs the correct answer. But we do take correct semantic dependency into comparison, which is usually weaker than wrong dependencies when the model makes mistakes.

---

> ### Author Response · Authors · 2024-11-25
>
> **Q4.2 The experimental methodology seems a bit circular. The intended conclusion is that for cases where the model chooses the incorrect answer, the incorrect answer has higher saliency towards the question tokens than the correct answer. This doesn’t necessarily seem to be surprising, and it’s not clear what is actionable about this finding.**
>
> We appreciate your feedback.
>
> A4.2.1 (Critical Gap) Given a specific QA task, Our study can explain why the model outputs wrong answers from **a semantic dependency perspective**. For example, in the case of a question like, “Where do A live?” with context stating, “A lives on an island. B lives in a mountain.” the model may incorrectly output “mountain” instead of “island.” We can examine the relationship between “mountain” and question tokens to see if they are falsely dependent. The prior study usually works by **analyzing model performance** on various types of adversarial examples and attributing the decline to broader issues, such as the model's tendency to rely on surface-level features like word overlap and positional cues. But we offer a finer-grained explanation for model errors.
>
> A4.2.2 (Potential Applications) We have discussed future model design in the Introduction. Future research could refine attention mechanisms to better prioritize meaningful
> token interactions. Also, the **semantic dependency between tokens can be localized** (This part is added to this section and appendix). We have designed a method based on token perturbation to localize the attention head groups that are responsible for the semantic dependency between tokens. We can also evaluate how much a single attention head contributes to a token-level semantic dependency. This provides insights for finetuning attention heads according to specific model errors.
>
> **5. [Nits] The terms “unfaithful” and “semantically dependent information” appear in the abstract without clear definitions. It wasn’t clear to me what their meaning was in this context. Perhaps this could be clarified to improve readability of the abstract.**
>
> We appreciate your suggestion. We will change the term "unfaithful" into “incorrect” and change the second sentence to “To understand the cause of this issue, we explore how semantic information is encoded and semantic dependency is learned within the model. And we will define "semantic dependency” and ”semantic information" in the Introduction to improve accessibility for readers as follows:
>
> **Semantic information** refers to the meaningful content that consists of data or representations that carry meaning interpretable in a specific context.
>
> **Semantic dependency** can be defined as the relationship between words in a sentence where the meaning of one word (predicate) depends on another word (argument) in the sentence.

---

> > ### Comment · Reviewer_mBPy · 2024-11-25
> >
> > Thank you for your response. I do think the paper would be improved with the proposed edits, in addition to the overall recommendations from the other reviewers. Perhaps with these revisions the paper may offer some interesting findings for an audience focused on interpretability and NLP. However, I will keep my original score, based on some of the factors mentioned in my original review.

---

> > > ### Author Response · Authors · 2024-11-27
> > >
> > > Thank you for your feedback. We have carefully addressed your concerns in our revised paper and response. To further improve the quality of our work, **we would greatly appreciate it if you could share the specific points or concerns that you believe require additional clarification or improvement**. We will provide further revisions and explanations.

---

### Official Review · Reviewer_DNKk · 2024-11-02

**Soundness:** 2
**Presentation:** 3
**Contribution:** 2
**Rating:** 6
**Confidence:** 4

**Summary:**

This paper delves into the internal mechanisms of transformer models to explore how semantic information is propagated and aggregated across tokens, which can contribute to the errors produced by large language models (LLMs). Experimental results under four settings have illustrated several useful findings.

**Strengths:**

1. The motivation and flow of this paper is clear.
2. Experimental results have demonstrated several useful findings.

**Weaknesses:**

1. Figure 1 is not illustrative enough. For example, in Figure 1c, it is hard to understand the key idea, why the two arrows are in the opposite direction? Why one sequence has green followings and the other does not? The answers cannot be found until reading line 112-120 (and still confusing).
2. Line 184: if the perturbation token is sampled randomly, it is possible that the semantic information of the original sentence does change a lot due to the perturbation token, meaning that the semantic information change may due to the dependency between this perturbation token and the other token in the original input. Thus, the authors may need to consider all the tokens in each sequence when calculating the average change of the jth token.
3. The experiment results in Table3 also reflects what I have mentioned in point 2.
4. The notations can be clearer.
5. It seems that only one NLP task (QA) has been investigated. It seems that wrongly aggregate information may affect tasks like reasoning and in-context learning. Maybe the authors can investigate more tasks to obtain more comprehensive results.

**Questions:**

Please see the above weakness section.

---

> ### Author Response · Authors · 2024-11-25
>
> **Q1. [Clarify Figure] Figure 1 is not illustrative enough. For example, in Figure 1c, it is hard to understand the key idea, why the two arrows are in the opposite direction? Why one sequence has green followings and the other does not? The answers cannot be found until reading line 112-120 (and still confusing).**
>
> Thank you for highlighting the need to improve Figure 1 and its explanation. We will ensure that the following points are explicitly clarified in the main text near the figure:
>
> A1.1 (Opposite Arrows in Figure 1c) The arrows represent the semantic information propagation flow from the **token “rhino” at layer 0** to a **group of tokens at layer L**. Tokens with darker blue backgrounds at layer L are more strongly related to the token “rhino.” Red highlights indicate tokens whose semantic dependency rankings relative to "rhino" change when input context changes. For instance, “white” becomes more strongly associated with “rhino” than “gray” when the irrelevant context “apples are red” is added. The arrows' opposite directions are designed to visually align the tokens at the final layer for easier comparison. However, we recognize this may be confusing, and in the revised version, we will provide a more detailed caption to explain this aspect clearly.
>
> A1.2 (the green followings) The green background highlights represent **irrelevant context** in the sequence. In this example, "apples are red" serves as irrelevant context to "white rhinos are grey." For the **context-change example** (left of Figure 1c), we compare the token “rhino” across two scenarios: one where “white rhinos are grey” appears alone and another where irrelevant context is added. This comparison illustrates how relationships between “rhino” and other tokens shift when irrelevant context is introduced. For the **order-change example** (right of Figure 1c), we change the irrelevant context position (highlighted with green background) but maintain overall  semantic information unchanged to show order change can still affect the token relationship in "white rhinos are grey".
>
> **Q2. [clarify our experiment] Line 184: if the perturbation token is sampled randomly, it is possible that the semantic information of the original sentence does change a lot due to the perturbation token, meaning that the semantic information change may due to the dependency between this perturbation token and the other token in the original input. Thus, the authors may need to consider all the tokens in each sequence when calculating the average change of the jth token.**
> (Experiment Setting)  We appreciate your concern and clarify our experimental setup: In our experiments, we do not limit the calculations to a few randomly sampled tokens. Instead, we compute changes for nearly all tokens (95%) in each sequence, excluding special tokens such as [CLS] and [SEP]. This ensures a comprehensive evaluation of semantic dependencies across the input sequence. We will revise this experiment detail to ensure readers understand that the analysis accounts for the full token sequence rather than a subset.
>
> **Q3. [clarify our experiment] The experiment results in Table3 also reflects what I have mentioned in point 2.**
>
> A3.1 (What Table 3 Show) Thank you for your concern. In Table 3, the results show that tokens are primarily influenced by semantically dependent tokens in different models, further supporting the robustness of our approach.
>
> A3.2 (Datasets Preparation) In Experiment 3, we utilize SpaCy to generate a specialized word dependency dataset derived from the GLUE dataset. This dataset includes sentences from the GLUE dataset, where each word (as 1 case) in the sentence is annotated with its semantically dependent word group as standard dependency data. For each sentence in the dataset, SpaCy generates a set of semantically dependent tokens for nearly all tokens (over 90%, except for some special tokens like punctuation tokens).
>
> A3.3 (Experiment Setting) Experiment 3 evaluates our method's alignment with the semantic dependencies in the above specialized dataset. Specifically, for each token variation, we calculated an alignment score against the standard dependency data to validate that the token behavior aligns with semantically dependent tokens. The high alignment scores demonstrate that our method reliably captures the influence of semantically dependent tokens.

---

> ### Author Response · Authors · 2024-11-25
>
> **Q4. [Annotations] Annotations The notations can be clearer.**
>
> In the revised version, we will improve the notations and add definitions to make them clearer. We will also add a symbol list in the appendix.
>
> **Q5. [question-answering tasks] It seems that only one NLP task (QA) has been investigated. It seems that wrongly aggregate information may affect tasks like reasoning and in-context learning. Maybe the authors can investigate more tasks to obtain more comprehensive results.**
>
> Thank you for the suggestion. Though the scope of our study is currently limited to question-answering (QA) tasks, this setting allows for a **controlled analysis** of how semantic information propagates through the model. We will incorporate explanations into the revised version to clarify our task selection rationale and outline potential directions for broader exploration.
>
> A5.1 (Choice of QA as the primary task) We chose the question-answering (QA) task because it is particularly well-suited for evaluating the impact of semantic dependency errors at the token level. QA tasks inherently involve understanding and associating tokens in a question with those in the context, making them ideal for testing the model's ability to handle complex dependencies. This directly aligns with the focus of our study, which explores how semantic dependencies lead to errors in aggregation.
>
> A5.2 (Need for Ground Truth Datasets) To validate our findings, it is crucial to have ground-truth datasets that clearly present correct and incorrect dependencies. QA tasks provide such datasets, where the answers are explicitly tied to certain context tokens. These datasets enable us to systematically evaluate how dependency errors between question and context tokens contribute to prediction errors.
>
> A5.3 (Applicability to Other Tasks)
>
> We acknowledge that wrongly aggregated dependencies may affect other NLP tasks, such as reasoning and in-context learning. In future work, we plan to investigate tasks where dependency relationships are less explicit, such as natural language inference and commonsense reasoning.

---

> > ### Comment · Reviewer_DNKk · 2024-11-25
> > **Thanks for the response.**
> >
> > Thanks the authors for the clarification. I have increased my score correspondingly.

---

> > > ### Author Response · Authors · 2024-11-27
> > >
> > > Thank you for your positive response and for increasing your score. We appreciate your constructive feedback and will continue refining our work.

---

### Official Review · Reviewer_cEUh · 2024-11-04

**Soundness:** 3
**Presentation:** 2
**Contribution:** 2
**Rating:** 5
**Confidence:** 4

**Summary:**

This paper examines where information is stored in transformer language model tokens, and how it moves through the layers. For example, how "white" gets attached to rhinos in "white rhinos". The authors use minimal pair counterfactuals to judge how changing one token in the input of the model changes the value in a later layer. The authors use this to show that individual tokens contain mostly their own info ('rhinos' contains 'rhinos' information) and when models make mistakes when making predictions about some token, it is because the wrong information from context gets attached to that token.

**Strengths:**

The authors outline several interesting research questions related to the movement of information within individual tokens and explain cases of irrelevant information moving into the representations of key tokens that cause incorrect answers. I think the questions are interesting and the problems being addressed have been observed for some time, although there has not been a principled study on movement of semantically dependent information across layers to my knowledge.

This work connects nicely to previous work on "attention is not explanation" (e.g.,  https://dl.acm.org/doi/10.1145/3447548.3467307 ), but i think it does not expand enough on this body of work

The authors consider a range of models, including encoder-only models like BERT, which I think is a strength for this kind of study

**Weaknesses:**

While this work is original, I'm not sure how much the findings weren't already understood to be the case. For example, distracting text causing performance drops is well known. The authors show that this is due to semantic information from other tokens being encoded in the correct token, but the methods that show the extent to which this is true do not explain enough about how this information is stored in the model. If we really understood how this was represented in (and intruding on) a given token, we would be able to remove it. This information belongs to some value vector(s) that copied the information in some attention operation. Can that be localized? I am generally positive about this paper, but I believe it could go a bit deeper in its analysis to better quantify why certain failure modes take hold.

The figures are unclear and sometimes poorly annotated. For example, figure 2 text should be much larger and it should be more apparent what "score" is. Figure 3, it is unclear what the numbers are next to the arrows

Typos and suggestions:

* L15: semantic dependency is redundant. I would suggest rewriting parts of the abstract to be a bit more descriptive

**Questions:**

Can MLPs introduce erroneous semantic information in the early layers? for example, see https://arxiv.org/abs/2304.14767

---

> ### Author Response · Authors · 2024-11-25
>
> **Q1. [Compare to Existing Work] While this work is original, I'm not sure how much the findings weren't already understood to be the case. For example, distracting text causing performance drops is well known.**
>
> Thank you for addressing this concern. We will add related works and clarify our new findings.
>
> A1.1 (Existing Work) Existing work indeed discusses the phenomenon that irrelevant context in prompts can lead to a significant decline in model accuracy. For example, Shi et al. (2024) demonstrates that the inclusion of irrelevant context in prompts can lead to erroneous focus on unrelated content, causing a significant decline in model accuracy.
>
> A1.2 (Difference with Ours) Our study tries to explain why and provide an underlying reason for the decline in model accuracy given irrelevant context in prompts. We found that the rank of different semantic information (concepts) encoded in a token changes dramatically with or without irrelevant context in prompts. Specifically, we show that the proportion of affected tokens’ semantic dependency is rather high (around 25%–40%) in different tested models.
>
> A1.3 (Our Extra Finding) Additionally, we also found that even an **order change of irrelevant context position** can still affect the rank of different semantic information encoded in tokens while maintaining overall semantic information unchanged.
>
> A1.4 (Importance) Our insight can further help in training or fine-tuning a robust language model in which the rank of encoded semantic dependency within tokens is stable when given irrelevant context in prompts.
>
> A1.5 (Our Change in Revision) We have also revised our related work accordingly as follows: Existing works (Shi et al., 2024) have demonstrated that the inclusion of irrelevant context in prompts can lead to a significant decline in model accuracy. Our study tries to explain why and provide an underlying reason for such a decline in model accuracy. We found that the rank of different semantic information (concepts) encoded in a token changes when adding irrelevant context or simply changing the order of the context sequence.
>
> **Q2. [Clarify Our Method] The authors show that this is due to semantic information from other tokens being encoded in the correct token, but the methods that show the extent to which this is true do not explain enough about how this information is stored in the model.**
>
> Thank you for addressing this concern.
>
> A2.1 The token perturbation method can show how much semantic information from each token in the input sequence is stored in each token. Our experiment in Section 4 shows that question tokens are likely encoded with more semantic information from wrong answer tokens than correct tokens when the model makes mistakes.
>
> A2.2 For a token’s semantic dependency, we also designed a method to localize how the model’s attention heads contribute to it. We will introduce this method and corresponding results in the appendix and continue studying precise information storage in our future work.
>
> **Q3. [Potential Applications for Removing Erroneous Information] If we really understood how this was represented in (and intruding on) a given token, we would be able to remove it. This information belongs to some value vector(s) that copied the information in some attention operation. Can that be localized? I am generally positive about this paper, but I believe it could go a bit deeper in its analysis to better quantify why certain failure modes take hold.**
>
> We appreciate the reviewer’s insightful comment on the potential to localize and remove erroneous information during attention operations. While we do not yet provide a complete solution for removing intrusive information, our current analysis offers a key step in this direction by **identifying specific attention heads responsible for the transmission of erroneous semantic information**. Our study is as follows:
>
> A3.1 **Semantic dependency can be localized**. We have designed a method based on token perturbation to localize the attention head groups that are responsible for the semantic dependency between tokens. We can also evaluate how much a single attention head contributes to a token-level semantic dependency.
>
> A3.2 However, we also found that the model’s attention head performance for semantic information storage differs in various QA cases, making it impossible to unify a group of specific heads for general mistakes. We will introduce this method and corresponding results in the appendix and continue studying precise information storage in our future work.

---

> ### Author Response · Authors · 2024-11-25
>
> **Q4. [Figures and Annotations] The figures are unclear and sometimes poorly annotated. For example, figure 2 text should be much larger and it should be more apparent what "score" is. Figure 3, it is unclear what the numbers are next to the arrows**
>
> Thank you for the suggestion. We agree that the figures could be improved for clarity. In the revised version, we will enlarge the text and labels for Figure 2. We will note the "Dependency Alteration Score (DAS)" in Figure 2 and provide context for the numbers in Figure 3 as “semantic dependency scores from context tokens to question tokens.” We will add captions that clearly describe the content and purpose of each figure.
>
> **Q5. [Abstract Revision] L15: semantic dependency is redundant. I would suggest rewriting parts of the abstract to be a bit more descriptive**
>
> Thank you for the suggestion. We will revise the abstract and provide a clear definition in the introduction for the term "semantic dependency.”
>
> **Q6. [Question] Can MLPs Introduce Erroneous Semantic Information in Early Layers?**
>
> Thank you for the question.
>
> A6.1 Our current work focuses on **attention layers** as the primary mechanism for token-level information propagation. However, we acknowledge that MLPs could play a role in introducing or amplifying erroneous semantic information.
>
> A6.2 (Our Change in Revision)
>
> We will add this to the discussion and future work in the later version as follows:
>
> Studies like Mor et al. (2024) (Dissecting Recall of Factual Associations in Auto-Regressive Language Models) have shown that MLPs encode enriched representations that propagate attributes. Such representations may inadvertently amplify irrelevant or erroneous semantic information. We aim to extend our analysis to quantify the contribution of MLPs to semantic dependency in the future.

---

### Official Review · Reviewer_QEqH · 2024-11-04

**Soundness:** 2
**Presentation:** 1
**Contribution:** 1
**Rating:** 3
**Confidence:** 3

**Summary:**

This paper studies how semantic information from a sequence is encoded and aggregated in a single token position. They find that the information initially contained in a token is mostly retained at that position, that how the context information is aggregated in a token can be influenced by irrelevant context, and how some mistakes can be attributed to incorrect information propagation in a token.

**Strengths:**

* The paper does its experiments on different models across several model families (BERT, Llama, GPT).
* The framework of understanding mistakes are caused by erroneous information propagation is interesting. While it is likely that not all mistakes are caused by this type of error, this mechanism seems to be a promising way of understanding how transformers can get answers wrong.

**Weaknesses:**

* The paper is hard to follow and read. For example, the key terms 'semantic information' and 'semantic dependence' is not defined. Semantic dependency seems to be defined later in section 3 at L273, but it seems to be a pretty general definition that makes categorising false dependencies in Section 4 based on a post-hoc 'dependency is false if the answer is wrong'.
* There seems to be not much applications or possible interventions based on these findings. For example, we already know that LMs can be distracted by irrelevant contexts or context changes [2].
* The claim of 'understanding mistakes' is too broad, as they study only question-answering tasks where the answer already exists verbatim in the context.
* The paper does not discuss any related works on interpreting how information flow in transformers to answer questions, such as [1].

[1] Dissecting Recall of Factual Associations in Auto-Regressive Language Models
[2] Large Language Models Are Not Robust Multiple Choice Selectors

**Questions:**

* In L303, are the 10,000 cases described a subset of the 600,000 token cases described in L249? If so, how were these cases chosen?
* Detecting errors require you to know which tokens are incorrect to begin with. Do you have any potential application of detecting potential QA errors at scale without prior knowledge of the ground truth?

---

> ### Author Response · Authors · 2024-11-25
>
> Thank you for the constructive comments on related work and definition.
>
> **Q1. [Clarify Definition] The key terms 'semantic information' and 'semantic dependence' is not defined. Semantic dependency seems to be defined later in section 3 at L273, but it seems to be a pretty general definition that makes categorising false dependencies in Section 4 based on a post-hoc 'dependency is false if the answer is wrong'.**
>
> We appreciate the reviewer’s feedback on the absence of key word definitions. We will add them to the Introduction section of the revised version.
>
> A1.1 **Semantic information** refers to the meaningful content that consists of data or representations carrying meaning interpretable in a specific context.
>
> A1.2 **Semantic dependency** can be defined as the relationship between words in a sentence where the meaning of one word (predicate) depends on another word (argument) in the sentence.
>
> A1.3 In our case, **false semantic dependency** means the meaning of one word is not dependent on another. For example, in the sequence “blue sky and red apple,” the semantic dependency between the word “blue” and “apple” is false. In the experiment of Section 4, we view the model’s wrong output token and the question token as a false dependency for evaluation.
>
> **Q2. [Applications or Possible Interventions] There seems to be not much applications or possible interventions based on these findings. For example, we already know that LMs can be distracted by irrelevant contexts or context changes**
>
> Thank you for raising this concern. We will add the related work and address the applications or possible interventions based on our new findings in the revised paper.
>
> A2.1 (Existing Work) Existing work indeed discusses the **phenomenon** that irrelevant context in prompts can lead to a significant decline in model accuracy. Specifically, Shi et al. (2024) demonstrates that the inclusion of irrelevant context in prompts can lead to an erroneous focus on unrelated content, causing a significant decline in model accuracy.
>
> A2.2 (Difference With Ours) Our study tries to explain why and **provides an underlying reason** for the decline in model accuracy given irrelevant context in prompts. We found that **the rank of different semantic information (concepts) encoded in a token** changes dramatically with or without irrelevant context in prompts. Specifically, we show that the proportion of affected tokens’ semantic dependencies is rather high (around 25%–40%) in different tested models.
>
> A2.3 (Our Extra Finding) Additionally, we also found that even **order changes of irrelevant context positions** can still affect the rank of different semantic information encoded in tokens while maintaining overall semantic information unchanged.
>
> A2.4 (Applications or Possible Interventions) We believe our insight can further help in training or fine-tuning a robust language model by constraining the rank of encoded semantic dependencies within tokens when given irrelevant context in prompts.
>
> **Q3. The claim of 'understanding mistakes' is too broad, as they study only question-answering tasks where the answer already exists verbatim in the context.**
>
> Thank you for raising this concern. We have acknowledged this point as a limitation and future work. We agree that the scope of our study is currently limited to QA tasks where the answer exists verbatim in the context. This setting allows for a controlled analysis of how semantic information propagates across known input tokens. We will further emphasize this point in the Experiment section.
>
> **Q4. [Related Works] The paper does not discuss any related works on interpreting how information flow in transformers to answer questions**
>
> We appreciate the suggestion to include related works, particularly those that focus on understanding information flow in transformers for QA tasks.
>
> (Our Change in Revision)
> In the revised version, we will include a related work section and discuss the paper you mentioned as follows:
>
> Mor et al. (2024) analyze how factual associations are recalled in auto-regressive language models, highlighting the roles of MLP sublayers in enriching subject representations and attention heads in extracting attributes. Our study addresses a gap by studying how semantic information flows between tokens through attention layers in both non-auto-regressive (BERT) and auto-regressive models (GPT, Llama).

---

> > ### Comment · Reviewer_QEqH · 2024-11-26
> >
> > I thank the authors for their response.
> >
> > 1. Applications and possible interventions
> >
> > It is true that existing work has found the phenomenon that models can be distracted by irrelevant context, while this work finds an internal mechanism in the transformers that correlates with this. I agree with reviewer's cEUh's perspective that we do not understand the mechanism well enough. The paper would greatly benefit from a deeper investigation into the mechanism. For example, can you do some intervention to correct the mistakes when the model does a QA task wrongly?
> >
> > 2. Tasks past QA
> >
> > I find that that the tasks only being QA where the answer exists in the context to be extremely limiting for the paper.
> >
> > ---
> >
> > Overall, after reading the other reviews and authors reponses, my thought process is as follows:
> >
> > (1) Does this paper show some surprising result about how transformers make mistakes? No, the failure modes are well understood externally, though the authors do show some internal mechanism that correlates with the failure modes.
> >
> > (2) If not, does the paper then convincingly show that the mechanism exists in a meaningful way and is causally responsible for certain behaviors of the model? Unfortunately, I don't think the paper has hit this bar. I would encourage the authors to do further work into understanding how this mechanism work and how significant this mechanism is in the model's functioning.

---

> > > ### Author Response · Authors · 2024-11-27
> > >
> > > Thank you for your feedback.
> > >
> > > **Q1. The paper would greatly benefit from a deeper investigation into the mechanism. For example, can you do some intervention to correct the mistakes when the model does a QA task wrongly?**
> > >
> > > While we do not yet provide a complete solution for removing intrusive information to correct model behavior, our current analysis offers a key step in this direction by **identifying specific attention heads responsible for the transmission of erroneous semantic information**. We have added this part in **Appendix A.4**. We can also evaluate how much a single attention head contributes to a token-level semantic dependency.
> > >
> > > **Q2. I find that the tasks only being QA where the answer exists in the context to be extremely limiting for the paper.**
> > >
> > > Thank you for your concern. Though the scope of our study is currently limited to question-answering (QA) tasks where the answer exists in the context, this setting allows for a **controlled analysis** of how semantic information propagates through the model. We have incorporated explanations into the revised version to clarify our task selection rationale and outline potential directions for broader exploration.
> > >
> > > **A2.1 (Choice of such QA as the primary task)** We chose the question-answering (QA) task where the answer exists in the context because it is particularly well-suited for evaluating the impact of semantic dependency errors at the token level. QA tasks inherently involve understanding and associating tokens in a question with those in the context, making them ideal for testing the model's ability to handle complex dependencies. This directly aligns with the focus of our study, which explores how semantic dependencies lead to errors in aggregation. Knowledge or contexts beyond inputs will be a distraction to our analysis.
> > >
> > > **A2.2 (Need for Ground Truth Datasets)** To validate our findings, it is crucial to have ground-truth datasets that clearly present correct and incorrect dependencies. Such QA tasks provide such datasets, where the answers are explicitly tied to certain context tokens. These datasets enable us to systematically evaluate how dependency errors between question and context tokens contribute to prediction errors.
> > >
> > > **A2.3 (Applicability to Other Tasks)** We acknowledge that wrongly aggregated dependencies may affect other NLP tasks, such as reasoning and in-context learning. In future work, we plan to investigate tasks where dependency relationships are less explicit, such as natural language inference and commonsense reasoning.
> > >
> > > **Q3. Does this paper show some surprising result about how transformers make mistakes? No, the failure modes are well understood externally, though the authors do show some internal mechanism that correlates with the failure modes.**
> > >
> > > A3.1 Regarding your point about the external understanding of failure modes, **we would appreciate it if you could provide studies that well understand externally the specific examples of failure modes**. We believe if we are unaware of how internal structures work, it would be difficult to optimize and understand the model effectively.
> > >
> > > A3.2 For example, in the case of a question like, “Where do A live?” with context stating, “A lives on an island. B lives in a mountain.” the model may incorrectly output “mountain” instead of “island.” Knowing the existence of irrelevant context is not enough to explain why it happens. In our work, we can examine the relationship between “mountain” and question tokens to see if they are falsely dependent. The **prior study usually works by analyzing model performance** on various types of adversarial examples and attributing the decline to broader issues, such as the model's tendency to rely on surface-level features like word overlap and positional cues. However, we offer a **finer-grained explanation** for model errors, which provides an evaluation method for various language models and insights into fixing these errors.
> > >
> > > **Q4. If not, does the paper then convincingly show that the mechanism exists in a meaningful way and is causally responsible for certain behaviors of the model? Unfortunately, I don't think the paper has hit this bar. I would encourage the authors to do further work into understanding how this mechanism work and how significant this mechanism is in the model's functioning.**
> > >
> > > A4.1 In experiments across a large number of QA cases and different models, we observed a very high percentage indicating a strong correlation between false semantic dependencies and model errors.
> > >
> > > A4.2 Our further exploration in Appendix A.4 locates attention heads responsible for any token-level semantic dependency, indicating semantic dependency is mutually contributed by a small group of attention heads. We also found that the model’s attention head performance for semantic information storage differs in various QA cases.

---

> ### Author Response · Authors · 2024-11-25
>
> **Q5. [Questions: Clarify Experimental Setting] In L303, are the 10,000 cases described a subset of the 600,000 token cases described in L249? If so, how were these cases chosen?**
>
> Thank you for your question. In the revised version, we will include a detailed explanation in the corresponding experiments to make it clear.
>
> A5.1 (Datasets Preparation) The 10,000 cases are not a direct subset of the 600,000 token cases from L249. Instead, these 10,000 cases were derived from a specialized word dependence dataset we generated using SpaCy. This dataset includes sentences from the GLUE dataset, where each word (as one case) in the sentence is annotated with its semantically dependent word groups as standard dependency data.
>
> A5.2 (Experiment Setting) The 10,000+ token variation cases evaluate our method's alignment with the semantic dependencies in this proxy ground-truth dataset. Specifically, for each token variation, we calculated an alignment score against the standard dependency data to validate that the token behavior aligns with semantically dependent tokens. This evaluation demonstrates that our method effectively captures the semantic dependencies between tokens.
>
> **Q6. [Questions: Detecting Potential QA Errors] Detecting errors require you to know which tokens are incorrect to begin with. Do you have any potential application of detecting potential QA errors at scale without prior knowledge of the ground truth?**
>
> Thank you for your question.
>
> A6.1 Note that the aim of our method is not to predict how likely the answer is wrong given a question. There are some **unsupervised** evaluation methods based on confidence scores for this aim, such as:
>
> [1] Muttenthaler et al. 2020. Unsupervised Evaluation for Question Answering with Transformers.
>
> [2] Deng et al. 2023. Characterizing Prediction Matrix for Unsupervised Accuracy Estimation.
>
> A6.2 The aim of our method is to explain the reason why the model makes mistakes in Q&A. Given a large number of Q&A datasets, we can effectively use them. Our detection method is also fully automatic and can run at scale.

---

### Official Review · Reviewer_iRzN · 2024-11-04

**Soundness:** 2
**Presentation:** 1
**Contribution:** 2
**Rating:** 3
**Confidence:** 4

**Summary:**

This paper investigates how the semantic dependency encoded by tokens change within the model architecture, and how it influences the prediction of the model. The authors discuss that (1) many tokens' semantics do not change with deeper layers; (2) a token also encode information of semantically related words; (3) changes of context, even irrelevant, can change the semantic dependency; (4) when models make mistakes, there are erroneous semantic dependency.

**Strengths:**

This paper tackles an important problem about understanding the model’s erroneous behavior by looking into the internal activations. The authors’ attempt to attribute the model’s prediction to semantic dependency can be a meaningful trial towards a more broad mechanistic understanding of the connections between internal activation and model behavior.

**Weaknesses:**

I feel the conclusions draw by the authors are in general discussed by existing literature and known by the overall language model community, and may not be strong enough to make a ICLR paper. Specifically:

- The conclusion that the model’s activation does not change much through layers is already observed by a thread of work, based on which there are techniques derived like early exist: Liao et. al. 2021, Schuster et. al. 2022, Elhoushi et. al. 2024.
- The conclusion that language models behavior changes with perturbation by irrelevant context is observed in Shi et. al. 2024.
- The conclusion that model mistakes can be attributed to incorrect dependency is also observed in Wu et. al. 2024.

Given the listed works, I tend to feel that this work is more or less describing the conclusions that are already aware by the community (though with their own metrics).

In addition, I find it difficult to read and navigate notations from line 265, page 5 to line 347, page 7, mostly because the authors use a large set of symbols without clearly state their definition and the motivation of using them. This part of the paper may require significant rewrite and clarification. I would suggest the authors may compile a list of symbols to clarify the meaning of all symbols and the motivation of using them, and maybe use diagrams and specific examples to illustrate the procedure described in Section 3.

References:

Liao et. al. 2021. A Global Past-Future Early Exit Method for Accelerating Inference of Pre-trained Language Models

Schuster et. al. 2022. Confident Adaptive Language Modeling

Elhoushi et. al. 2024. LayerSkip: Enabling Early Exit Inference and Self-Speculative

Shi et. al. 2024. DecodingLarge Language Models Can Be Easily Distracted by Irrelevant Context

Wu et. al. 2024. Retrieval Head Mechanistically Explains Long-Context Factuality

**Questions:**

See above weakness section.

---

> ### Author Response · Authors · 2024-11-25
> **Response by Authors**
>
> Thank you for the constructive comments regarding the related work. Our work is very different from the existing one. We have added your mentioned work into our related work to make it more comprehensive and avoid confusion.
>
> **Q1. [first finding overlap] The conclusion that the model’s activation does not change much through layers is already observed by a thread of work, based on which there are techniques derived like early exist: Liao et. al. 2021, Schuster et. al. 2022, Elhoushi et. al. 2024.**
>
> A1.1 (Existing work) Existing work, including Liao et al. (2021), Schuster et al. (2022), and Elhoushi et al. (2024), focuses on **model activation stability** in the later layers of transformer models. Specifically, later layers may contribute minimally to the refinement of token representations, which enables techniques like **early exit** to accelerate inference.
>
> A1.2 (Our finding) Our finding is that whether the $i$-th token of the last layer $L$ mostly contains its original semantic information from the $i$-th token in layer 0. This has not been studied by existing works.
>
> A1.3 (importance) The finding that the last-layer token retains its original semantic information is critical for multiple reasons. It ensures that semantic dependency analyses in later layers are meaningful and reliable. It also provides a stable basis for interpretability, enabling attention-based methods to trace token contributions accurately. Additionally, this property could inspire new architectures that further optimize semantic stability.
>
> A1.4 (our change in revision) We follow your constructive comments, and we revise the related work to make a clear difference with existing work as follows:
> Existing work (Liao et al., 2021; Schuster et al., 2022; Elhoushi et al., 2024) has studied model activation stability in **later layers** of transformer models. Specifically, additional layers may contribute minimally to the refinement of token representations, which enables techniques like early exit to accelerate inference. However, whether the $i$-th token of the last layer $L$ mostly contains its original semantic information from the $i$-th token in layer 0 has not been studied. Our finding ensures that semantic dependency analyses are meaningful and reliable. The fact that most tokens still predominantly reflect their initial semantics highlights the model's strong retention property, which is not inherently expected given the iterative aggregation of semantic information across many layers.
>
> **Q2. [second finding overlap] The conclusion that language models behavior changes with perturbation by irrelevant context is observed in Shi et. al. 2024.**
>
> A2.1 (Existing work) Existing work discusses the **phenomenon** that irrelevant context in prompts can lead to a significant decline in model accuracy. Specifically, Shi et al. (2024) demonstrates that the inclusion of irrelevant context in prompts can lead to an erroneous focus on unrelated content, causing a significant decline in model accuracy.
>
> A2.2 (difference with ours) Our study tries to explain why and **provides an underlying reason** for the decline in model accuracy given irrelevant context in prompts. We found that **the rank of different semantic information (concepts) encoded in a token** changes dramatically with or without irrelevant context in prompts. Specifically, we show that the proportion of affected tokens’ semantic dependency is rather high (around 25%~40%) in different tested models.
>
> A2.3 (our extra finding) Additionally, we also found even **order changes of irrelevant context positions** can still affect the rank of different semantic information encoded in tokens while maintaining overall semantic information unchanged.
>
> A2.4 (importance) We believe our insight can further help training or fine-tuning a robust language model in which the rank of encoded semantic dependency within tokens is stable when given irrelevant context in prompts.
>
> A2.5 (our change in revision) We have also revised our related work accordingly as follows: Existing works (Shi et al., 2024) have demonstrated that the inclusion of irrelevant context in prompts can lead to a significant decline in model accuracy. Our study tries to explain why and provide an underlying reason for such a decline in model accuracy. We found the rank of different semantic information (concepts) encoded in a token changes when adding irrelevant context or simply changing the order of the context sequence.

---

> ### Author Response · Authors · 2024-11-25
>
> **Q3. [third finding overlap] The conclusion that model mistakes can be attributed to incorrect dependency is also observed in Wu et. al. 2024.**
>
> A3.1 (Existing work) This work is parallel with our work. The work shows that **some attention heads** contribute largely to model performance, as they retrieve factual information. In other words, some parameters are important and help encode factual information. The absence or malfunctioning of these retrieval heads may lead to model errors.
>
> A3.2 (Our finding) Our study explores how token-level **semantic dependency** influences model mistakes. Specifically, wrong tokens are likely to have stronger semantic dependency on question tokens than correct tokens when models make mistakes.
>
> A3.3 (importance) Our findings provide another perspective on understanding and correcting model mistakes under specific question-answering cases.
>
> A3.4 (our change in revision) Thank you for your comments, we realize that we should add more work on attention head into our related work. Specifically, we revised our related work as follows:
>
> Existing works have studied specific roles of attention heads to explain model errors. Wu et al. (2024) identify specific attention heads, termed retrieval heads, which are critical for retrieving factual information from long contexts. The absence or malfunctioning of these retrieval heads may lead to model errors. Gandelsman et al. (2024) show some attention heads in CLIP have property-specific roles (e.g., location or shape), which are important for model performance. Our study addresses another reason by exploring how token-level semantic dependency influences model mistakes, which provides another critical perspective on understanding and correcting model mistakes under specific question-answering cases.
>
> References:
> Gandelsman et al. 2024. Interpreting CLIP's Image Representation via Text-Based Decomposition
>
> **Q4. rewriting**
>
> We appreciate the reviewer’s feedback on the complexity of notations. In the revised version, we have revised annotations and added a **symbol list in the appendix (page 19)** to make them comprehensible.

---

### Meta-Review · Area_Chair_rrZ9 · 2024-12-19

**Metareview:**

This paper studies how semantic dependency is captured in pretrained LMs. They find that many tokens do not change substantially over layers, and that analyzing this phenomena can be diagnostic of some of the errors made by models.

Despite some interesting analyses, there are major weaknesses in the paper with regard to novelty (many of the findings are already known), as well as clarity/writing. I am therefore recommending that this paper be rejected.

**Additional Comments On Reviewer Discussion:**

Many reviewers found the results not so novel in light of the large body of existing work in this area. Moreover, there was general consensus that the writing was unclear (e.g., no definition of "semantic dependency"), and the paper could be structured better. While the authors responded to such points in the rebuttal, the rebuttal did not change my opinion.

---

### Decision · Program_Chairs · 2025-01-22

Reject